**Data Availability Statement:** Bird and tree data used in this analysis are available through the

# Assessing trends and vulnerabilities in the mutualism between whitebark pine *(Pinus albicaulis)* and Clark's nutcracker *(Nucifraga columbiana)* in national parks of the Sierra-Cascade region

Chris Ray[ID][1]*, Regina M. Rochefort[ID][2], Jason I. Ransom[2], Jonathan C. B. Nesmith[3], Sylvia A. Haultain[3], Taza D. Schaming[4], John R. Boetsch[5], Mandy L. Holmgren[1], Robert L. Wilkerson[1], Rodney B. Siegel[1]

1 The Institute for Bird Populations, Petaluma, California, United States of America, 2 North Cascades National Park Service Complex, Sedro-Woolley, Washington, United States of America, 3 National Park Service, Sierra Nevada Network, Three Rivers, California, United States of America, 4 Northern Rockies Conservation Cooperative, Jackson, Wyoming, United States of America, 5 National Park Service, North Coast and Cascades Network, Port Angeles, Washington, United States of America

* cray@birdpop.org

## Abstract

Dispersal of whitebark pine (*Pinus albicaulis* Engelm.), a keystone species of many high-elevation ecosystems in western North America, depends on Clark's nutcracker (*Nucifraga columbiana* Wilson), a seed-caching bird with an affinity for whitebark seeds. To the extent that this dependence is mutual, declines in whitebark seed production could cause declines in nutcracker abundance. Whitebark pine is in decline across much of its range due to inter-acting stressors, including the non-native pathogen white pine blister rust (*Cronartium ribicola* J. C. Fisch.). We used avian point-count data and tree surveys from four national park units to investigate whether trends in whitebark pine can explain trends in Clark's nut-cracker. Spatial trends were modeled using recent data from two parks, while temporal trends were modeled using longer time-series of nutcracker and whitebark data from two additional parks. To assess the potential dependence of nutcrackers on whitebark, we linked a model of nutcracker density (accounting for detection probability) with a model of whitebark trends, using a Bayesian framework to translate uncertainty in whitebark metrics to uncertainty in nutcracker density. In Mount Rainier National Park, temporal models showed dramatic declines in nutcracker density concurrent with significant increases in whitebark crown mortality and trees infected with white pine blister rust. However, nutcrack-ers did not trend with whitebark metrics in North Cascades National Park Service Complex. In spatial models of data from Yosemite National Park and Sequoia-Kings Canyon National Park, nutcracker density varied not only with local cover of whitebark but also with elevation and, in Sequoia-Kings Canyon, with cover of another species of white pine. Our results add support for the hypothesis that the mutualism between whitebark pine and Clark's

National Park Service Data Store at https://irma.
nps.gov/DataStore/Reference/Profile/2277917 (for
NCCN parks data) and at https://irma.nps.gov/
DataStore/Reference/Profile/2278594 (for SIEN
parks data).

**Funding:** This analysis was funded by the National
Park Service Inventory & Monitoring Division
(www.nps.gov/im) and North Coast and Cascades
Research Learning Center (www.nps.gov/rlc/
northcoastcascades), and was conducted under
California Cooperative Ecosystem Studies Unit
Agreement P17AC01613 with The Institute for Bird
Populations (CR, MLH, RLW, RBS). Additional and
in-kind support was provided by Mount Rainier
National Park, North Cascades National Park
Service Complex, Olympic National Park, Sequoia
and Kings Canyon National Parks and Yosemite
National Park. This research was sponsored by the
National Park Service, personnel of which were co-
authors who collaborated on elements of the study
design related to data collection and analysis,
decision to publish and preparation of the
manuscript.

**Competing interests:** The authors have declared
that no competing interests exist.

nutcracker is vulnerable to disruption by blister rust, and our approach integrates data across monitoring programs to explore trends in species interactions.

## Introduction

Mutualism involves beneficial interactions among species in a partnership, creating the potential for feedback processes that result in mutual population growth or decline [1, 2]. Even when a mutualism is facultative (not obligatory) for one or more species in a partnership, population decline in one partner can lead to decline in another [3]. Accordingly, many species involved in mutualistic interactions are vulnerable to novel stressors, such as climate change or introduced pathogens, through the impact of those stressors on partner species, in addition to any direct impacts they might experience [4]. For example, ant-aphid mutualisms in which there are indirect, ant-mediated effects of climate on aphid population growth and behavior, can make ant-dependent aphids relatively sensitive to climate change [5]. Other examples of stress exacerbated by mutualism include plant-animal mutualisms in Hawaii, where habitat destruction contributed to the loss of bird species and associated population declines in many bird-pollinated plants [6]; and drought-stressed *Ficus* species in Borneo that experienced prolonged shifts in their reproductive phenology, leading to local extinction of the wasp that pollinated their figs [7]. The potential for a stressor to affect multiple species in a mutualism should scale with its potential to affect each species directly, indirectly through the mutualism, or both. Furthermore, the strength of a stressor's indirect effects should help reveal patterns of dependence in the mutualism. Here, we explore the dependence of a facultative mutualist on its obligate partner, by modeling the indirect effect of a pathogen. Specifically, we examine whether trends in Clark's nutcracker (*Nucifraga columbiana* Wilson), a seed-caching bird species, can be explained by pathogen-mediated trends in whitebark pine (*Pinus albicaulis* Engelm.), a tree that depends on Clark's nutcracker for seed dispersal and germination [8–11].

Whitebark pine (hereafter, whitebark) is a keystone species in high-elevation areas of western North America, influencing biodiversity, ecosystem structure, and hydrologic cycling [12, 13]. Over the past century, precipitous and widespread declines in whitebark survival and recruitment have been documented across much of the species' range [14–18]. Threats to whitebark include direct and interactive effects of attacks by an exotic fungal pathogen and a native insect pest, as well as climate change and fire exclusion [19]. White pine blister rust (hereafter, blister rust) is caused by a fungal pathogen (*Cronartium ribicola* J. C. Fisch) that was first recorded in North America in 1910 [20, 21]. Blister rust girdles the branches and boles of five-needle white pines, reducing cone production and causing up to 90% mortality in some parts of the whitebark range [21–23]. Additional whitebark mortality caused by outbreaks of the native mountain pine beetle (*Dendroctonus ponderosa* Hopkins) is facilitated by an increase in growing degree days [24, 25]. In 2004 alone, nearly 720,000 whitebark pines were killed by mountain pine beetles in the Greater Yellowstone Ecosystem [26].

Climate change might threaten whitebark directly through effects on growth, mortality and regeneration, as well as indirectly through increasing frequency, intensity and duration of impacts by blister rust, mountain pine beetle, and fire [27–29]. Projected effects of climate are complicated by the fact that whitebark benefits from early successional conditions [19]—conditions that would spread with increasing wildfire. On the other hand, in much of the species range, decades of fire exclusion have facilitated succession of whitebark communities by more shade-tolerant species, and have increased fuel accumulation, leading to larger and more severe fires that kill more trees (including potentially rust-resistant trees that have eluded rust-

mediated mortality) [14]. Given the uncertain future of whitebark, it is especially important to understand the potential for feedback effects to exacerbate whitebark loss.

Whitebark requires Clark's nutcrackers (hereafter, nutcrackers) to disperse its wingless seeds [8, 10]. Nutcrackers use a specialized sublingual pouch to move seeds—sometimes many kilometers—and often cache seeds at depths conducive to germination [9–11]. Although whitebark and nutcrackers are regarded as coevolved mutualists [9, 30], this mutualism can be facultative from the perspective of the nutcracker, which also forages on other pine species [8, 31, 32]. This facultative foraging appears to include active evaluation of cone production [33, 34], and nutcrackers will emigrate from areas where cone production falls below a minimum threshold [23]. Recent evidence suggests the decline of whitebark is leading to local declines in nutcracker populations [23, 35, 36]. Due to the obligate dependence of whitebark on nutcrackers, lower nutcracker density could reduce whitebark recruitment, and this positive feedback loop could lead to local declines in both species.

Both whitebark and nutcrackers have been selected for monitoring by the National Park Service (NPS) as part of the long-term "Vital Signs" program for identifying trends in natural resources on NPS lands [37]. Several NPS networks monitor whitebark and/or nutcrackers [38, 39], and both species are monitored in several parks, including Mount Rainier National Park (MORA) and North Cascades National Park Service Complex (NOCA) in the North Coast and Cascades Inventory and Monitoring Network (NCCN), as well as Yosemite National Park (YOSE) and Sequoia and Kings Canyon National Parks (SEKI) in the Sierra Nevada Inventory and Monitoring Network (SIEN). Avian surveys are conducted annually in all four parks using a shared protocol that produces data suitable for modeling the spatial and temporal dynamics of many landbird species [39]. Tree surveys conducted in these parks also share features key to modeling the potential dependence of nutcrackers on this resource, including live tree density, cone production and metrics of disease caused by blister rust [40, 41]. Although NPS Vital Signs monitoring is conducted within a coordinated framework [37, 42], relating data between vital signs is complicated by the fact that tree and avian survey plots were not co-located within parks (Fig 1).

Our objective was to integrate data from the separate surveys of whitebark and nutcrackers in these parks, to assess whether trends in nutcracker population density can be explained by trends in whitebark. In NCCN parks, tree surveys began in 2004 and avian surveys began in 2005. In SIEN parks, tree and avian surveys began in 2011. We used the longer series (2004–2016) of data from NCCN parks to model the temporal dynamics of nutcracker density as a function of whitebark metrics, and the shorter series (2011–2016) from SIEN parks to model spatial trends in nutcracker density. Models of whitebark and nutcracker dynamics were linked in a Bayesian framework to allow for uncertainty in whitebark metrics to affect uncertainty in nutcracker density. By allowing for uncertainty at both trophic levels, this model can be fit to data from distinct monitoring efforts to analyze ecological interactions such as mutualism and—our focus here—the potential dependence of nutcracker density on metrics of whitebark abundance, health and productivity.

## Methods

### Study areas and survey data

**Tree surveys.** In NCCN parks, whitebark occurs in relatively disjunct stands, including 66 stands distributed over ~1175 ha in the northeast quadrant of MORA and 12 stands distributed over ~3800 ha in the southeast portion of NOCA [40]. A random subset of these stands was monitored during 2004–2016, including eight stands in MORA and five stands in NOCA (Table 1). Each stand was surveyed within 2–7 permanent, randomly located, circular plots of

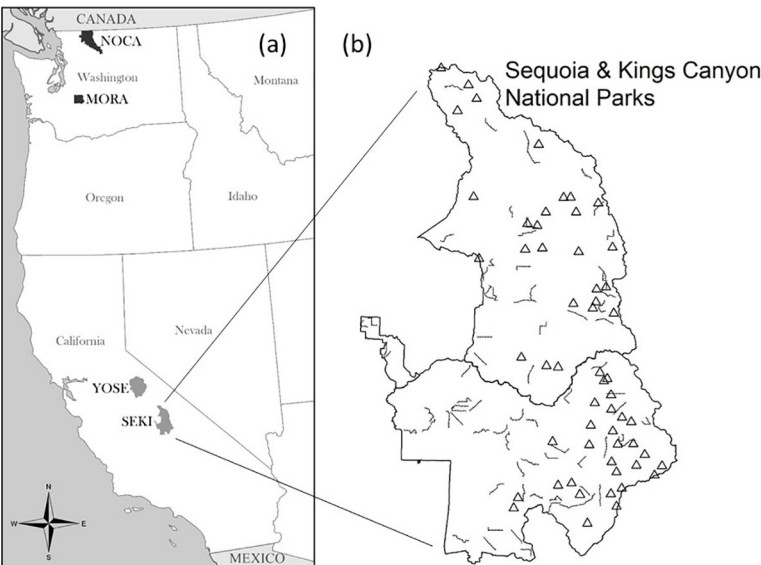

**Fig 1. National park units monitored for this study.** (a) Whitebark pine and Clark's nutcracker were monitored in Mount Rainier (MORA), North Cascades (NOCA), Yosemite (YOSE) and Sequoia and Kings Canyon (SEKI) national parks. Darker shading indicates parks in the North Coast and Cascades Inventory and Monitoring Network (NCCN) and lighter shading indicates parks in the Sierra Nevada Inventory and Monitoring Network (SIEN). (b) An example of the typically non-overlapping distribution of avian point-count transects (lines) and tree survey plots (triangles) within a park (here, SEKI).

0.04 ha each (n = 29 total plots in MORA, 35 in NOCA), and each plot was surveyed in multiple study years, resulting in 4–5 years of tree survey data from each park (Table 1).

In SIEN parks, whitebark serves as a foundational species in upper subalpine and treeline forests within YOSE and the northern half of SEKI [38, 41]. In the southern half of SEKI, however, whitebark is replaced as the dominant subalpine conifer by foxtail pine (*Pinus balfouriana* Balf.). Foxtail pine (hereafter, foxtail) is another five-needle white pine used as forage by nutcrackers [32]. SIEN plot surveys were distributed among three sampling frames: whitebark in YOSE, whitebark in SEKI, and foxtail in SEKI. Random plot locations within each target population were selected using a spatially balanced Generalized Random-Tessellation

**Table 1. Whitebark pine surveys conducted in Mount Rainier National Park (MORA) and North Cascades National Park Complex (NOCA), park units within the North Coast and Cascades Inventory and Monitoring Network (NCCN).** These data were used for temporal analyses.

| | | | Surveyed | | |
|---|---|---|---|---|---|
| Park | Year | Survey months | Plots | Trees[a] | Stands |
| MORA | 2004 | Jul—Sep | 29 | 237 | 8 |
| | 2007 | Jul—Sep | 9 | 54 | 3 |
| | 2009 | Aug—Sep | 29 | 251 | 8 |
| | 2015 | Aug—Sep | 29 | 265 | 8 |
| NOCA | 2004 | Jul—Sep | 35 | 255 | 5 |
| | 2007 | Aug | 14 | 188 | 2 |
| | 2009 | Jul—Sep | 35 | 265 | 5 |
| | 2015 | Sep—Oct | 21 | 238 | 3 |
| | 2016 | Jul—Aug | 14 | 59 | 2 |

[a]Counts include both live and dead whitebark surveyed.

**Table 2. Whitebark and foxtail pine surveys conducted in Yosemite National Park (YOSE) and Sequoia and Kings Canyon National Parks (SEKI), both within the Sierra Nevada Inventory and Monitoring Network (SIEN).** These data were used for spatial analyses.

| Park | Year | Survey months | Surveyed | | |
| | | | Plots | Whitebark[a] | Foxtail[a,b] |
|---|---|---|---|---|---|
| YOSE | 2011 | Jul—Aug | 11 | 1039 | - |
| | 2013 | Jul | 6 | 552 | - |
| | 2014 | Jun—Jul | 10 | 1246 | - |
| | 2015 | Jun—Aug | 12 | 1931 | - |
| | 2016 | Jun—Jul | 12 | 1039 | - |
| SEKI | 2011 | Aug | 7 | 86 | 225 |
| | 2013 | Jun—Sep | 14 | 628 | 446 |
| | 2014 | Jul—Sep | 18 | 1333 | 256 |
| | 2015 | Jul—Sep | 12 | 773 | 334 |
| | 2016 | Jun—Oct | 21 | 1328 | 549 |

[a]Counts include both live and dead trees surveyed.

[b]Foxtail do not occur in YOSE.

Stratified (GRTS) equal-probability sampling algorithm [43] to select 99 permanent plots across the three sampling frames (Table 2). Each 50x50-m plot was assigned to one of three serially-alternating panels for survey every three years. The mean number of surveys per plot during 2011–2016 was 1.68.

In both NCCN and SIEN tree surveys [40, 41], individual trees were identified to species, measured for diameter at breast height (to estimate *basal area*) and examined for status (live or dead), presence of female cones (*cone trees*), percent *crownkill* (a visual estimate of canopy mortality ranging 0–100), occurrence of mountain pine beetle, and blister rust infection. Signs of blister rust infection [44], often flagged by dead needles or branches, included active cankers (swollen and/or discolored sections of branches or boles), inactive cankers (rough or de-barked sections damaged by fruiting of the rust), and de-barked/oozing areas damaged by animals that have browsed on the sugary tissues associated with cankers.

**Avian surveys.** Multi-species avian point-counts were conducted within each park along a set of permanent transects distributed via GRTS algorithm among three strata: high, intermediate and low elevations [45, 46]. High-elevation transects originated above a threshold elevation for sampling subalpine and alpine habitats in each park (1350 m in NCCN parks, 2750 m in YOSE and 3000 m in SEKI). Transects were grouped into six panels, with each panel containing a balanced sample from all three strata. Every year, panel 1 was surveyed in each park, along with one of the remaining panels, in a serially alternating design (panel 1 and 2 in year *t*, 1 and 3 in year *t*+1, and so on). Along each transect, point-count stations were located at intervals of 200 m (NCCN) or 250 m (SIEN). Point counts were conducted from late May to late July of each year (2005–2016), with transects at higher elevations surveyed later in the season to align with the nesting phenology of many bird species. Conducting surveys later at higher elevations also accommodates the annual phenology of nutcracker foraging movements [8, 34, 47, 48].

At each point-count station, a trained observer recorded nutcracker detections and survey covariates according to protocol [45, 46]. Any nutcracker heard or seen during a seven-minute survey was recorded, along with its detection distance (meters from observer) and detection-time interval (0:00–3:00, 3:01–5:00 or 5:01–7:00 minutes), to support analyses that account for birds present but undetected [49–51]. Survey covariates included *observer*, *date*, *hour*, ambient *noise* level (categorized as 1 = low to 5 = high), presence of *forest* cover, and presence of *dense*

*cover* of any vegetation in which birds might escape detection. Covariates calculated from a digital elevation model using verified point-count station coordinates included *elevation*, *slope* angle and slope *aspect*.

Because variation in snow cover and the timing of spring snowmelt might affect cone production and nutcracker foraging behavior [33, 52], we calculated mean spring temperature (*MST*, mean daily temperature from March 1 through May 31) and annual precipitation-as-snow (*PAS*, millimeters of snow falling between August 1 and July 31) as potential covariates. We used ClimateWNA [53] as a source of downscaled climate data, accessing annual data and 1971–2000 normals from http://www.climatewna.com to calculate annual *MST* and *PAS* as anomalies for each point-count station [54]. For surveys in year *t*, we expected a lagged response of cone production to *MST* and *PAS* in year *t*-1, so we calculated lag-1 *MST* as the mean temperature anomaly from March 1 to May 31 of year *t*-1, and lag-1 *PAS* as the snowfall anomaly from Aug 1 of year *t*-2 to July 31 of year *t*-1. Finally, because *MST* and *PAS* generally covary, we avoided collinearity in our models by replacing *MST* with *rMST*, the residuals of a linear regression of *MST* on *PAS* [55].

**Permitting.** All fieldwork was reviewed through the National Park Service Scientific Research Permit and Reporting System, and was approved as compliant with applicable laws and policies including the National Environmental Policy Act and National Historic Preservation Act. Fieldwork in designated Wilderness was also approved as compatible with wilderness character (natural quality, untrammeled quality, undeveloped quality, and solitude or primitive and unconfined recreation).

## Analyses

We modeled trends in nutcracker density using a hierarchical model developed for evaluating covariates of avian abundance and detection probability [51]. This model integrates separate processes for bird abundance *N*, availability for detection $p_a$, and detectability $p_d$, while accommodating covariates on each process. The population process is defined by a Poisson model of *N* as a function of environmental covariates. The observation process defines bird count *y* as a binomial random variable determined by $p_d$ and *n*, the number of birds available for detection, while *n* is a binomial random variable determined by $p_a$ and *N*. In turn, $p_a$ is determined by *q*, the per-minute probability of non-detection, while $p_d$ is determined by *σ*, the scale parameter for the half-normal distribution of detection distances. Both *q* and *σ* can be modeled as responses to environmental covariates affecting detection, such as date, hour and observer. We specify each sub-model below, and extend the model to allow for nutcracker dependence on whitebark by including a proxy of whitebark seed production, *W*, as a modeled covariate of *N*. We considered several whitebark seed proxies, as well as one proxy of foxtail seed production, *F* (Table 3).

Our temporal analysis began with two hypotheses: H1) nutcracker density has varied through time independently of whitebark dynamics, and H2) temporal variation in nutcracker density has been influenced by whitebark dynamics. Alternative models of temporal variation in *N* were constructed by including either an effect of *year* (H1) or an effect of *W* (H2), our proxy for whitebark seed production (Table 3). Because *W* was measured only intermittently (Table 1), we modeled missing values by regressing measured values on *year*. Modeling *W* allowed us to model the annual variation in nutcracker density according to H2 (Fig 2). The directed acyclic graph in Fig 2 demonstrates how observed data (squares) were used to inform estimated parameters (circles) throughout our model. *W* includes both observed and estimated values, and provides a link between our models of whitebark (gray box) and nutcracker abundance (black box).

**Table 3. Proxies of seed production by whitebark (*W*) or foxtail (*F*) pine, and their hypothesized effects on Clark's nutcracker density in national parks.**

| Species code | Seed proxy | Effect | Definition |
|---|---|---|---|
| $W_t$ | *trees* | + | Temporal index[a] of live whitebark abundance within a park |
| $W_t$ | *canker trees* | - | Temporal index of live whitebark with active blister-rust cankers within a park |
| $W_t$ | *rust trees* | - | Temporal index of live whitebark with cankers or other signs of blister rust infection within a park |
| $W_t$ | *crownkill* | - | Temporal index of whitebark crown mortality within a park |
| $W_k$ | *whitebark cover*[b] | + | Spatial index[b] of whitebark cover within 125 m of each avian point-count station $k$ |
| $F_k$ | *foxtail cover* | + | Spatial index of foxtail cover within 125 m of each avian point-count station $k$ |

[a]Time-varying indices (subscript $t$) were used in temporal models of 2005–2016 nutcracker data.
[b]Spatial indices (subscript $k$) were used in spatial models of 2011–2016 nutcracker data.

The observation model (dashed box in Fig 2, detailed in 49–51] used data on time-to-first-detection, *j*, and distance-to-detection, *b*, to estimate detection probability based on two common assumptions: birds detected sooner are more available for detection (make sound or move more often), and birds detected at greater distances are more detectable (easier to hear or see). We first related detection interval *j* to *q*, the per-minute probability that a bird will not be detected, which determines $p_a$ [39, 51, 56]. We then used the common approach of approximating the decline in detection probability with distance from the observer using a half-normal distribution fitted to *b*, our (binned) data on distance-to-detection. Using this approach, $p_d$ was a function of $\sigma$, the fitted scale parameter of the half-normal distribution [39, 51, 57].

For temporal analyses of nutcracker counts in NCCN parks, the expected value of *N* in each year was estimated as the response variable in a generalized linear model (GLM), with either *year* or (modeled) *W* as a covariate. For spatial analyses of nutcracker counts in SIEN parks, we modified the model in Fig 2 by replacing the temporal model of *W* (gray box) with spatially referenced data on whitebark and foxtail cover extracted from park vegetation maps [58, 59], assuming no temporal variation (Table 3). The expected value of *N* at each point-count station was estimated using a GLM based on *W*, *W*×*F*, or other spatially referenced covariates.

For both spatial and temporal analyses, we used an *N*-mixture model to integrate observation and population processes [60], linking sub-models describing *q* and $\sigma$ of the nutcracker observation process with the sub-model describing expected nutcracker abundance, $\lambda_N$. We then extended this model by linking $\lambda_N$ to a GLM describing the expected abundance of our whitebark metric, $\lambda_W$ (gray box in Fig 2).

Specifically, the per-minute probability of non-detection at station *k* in year *t* was modeled as a logistic response to potential covariates of individual availability for detection, as

$$\text{logit}(q_{kt}) = \beta_{q0} + \boldsymbol{\beta}_q \boldsymbol{x}_{qkt}, \tag{1}$$

where the vector of candidate covariates $\boldsymbol{x}_q$ is listed in Table 4. Similarly, the scale parameter of the half-normal distribution describing the decline in detection probability with distance was modeled as a log-linear function of covariates $\boldsymbol{x}_\sigma$ (Table 4),

$$\log(\sigma_{kt}) = \log(\sigma_0) + \boldsymbol{\beta}_\sigma \boldsymbol{x}_{\sigma kt}. \tag{2}$$

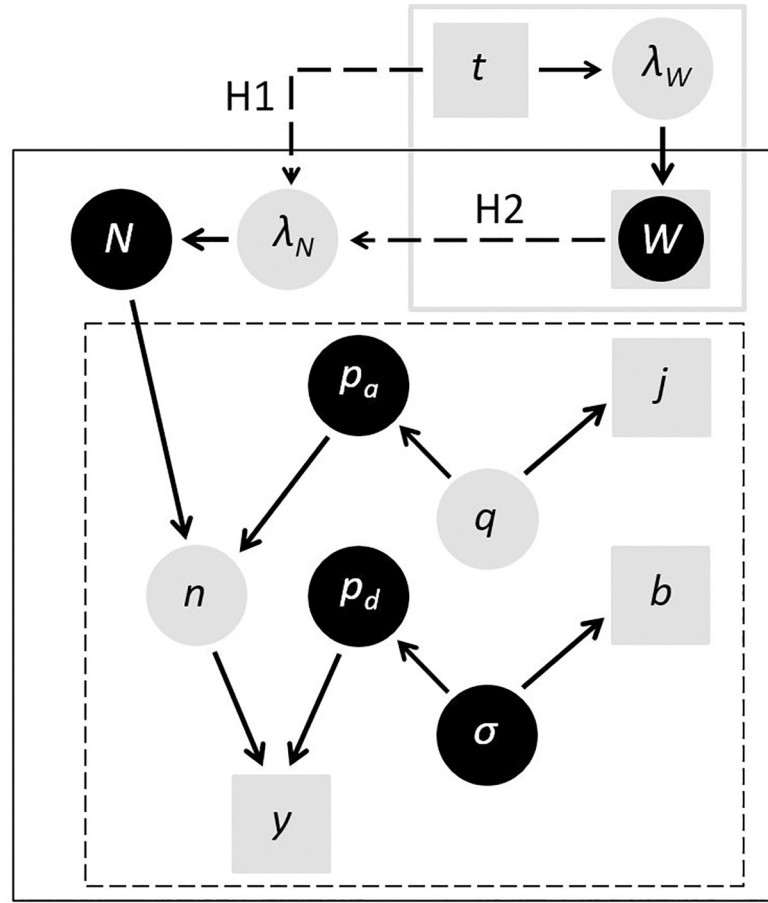

**Fig 2. Linked models of whitebark pine (gray box) and Clark's nutcracker (solid box), including details of the nutcracker observation model (dashed box).** Arrows depict assumed (solid) or hypothesized (dashed) dependencies between data (squares) and estimated parameters (circles), including focal parameters (dark circles), where $t$ = time (year), $\lambda_W$ = expected value of a whitebark metric (e.g., number of live trees), $W$ = realized whitebark metric (measured in some years), $\lambda_N$ = expected nutcracker abundance, $N$ = realized nutcracker abundance, $n$ = number of nutcrackers available for detection, $q$ = $1 - a$ = $1 -$probability of detection per minute, $p_a$ = probability of availability for detection during a count, $\sigma$ = scale of the half-normal function describing detection probability by distance, $p_d$ = probability of detection during a count, $y$ = nutcracker count (number detected), $j$ = time interval of detection, and $b$ = detection distance bin.

Data exploration (S1 File) suggested covariates likely to influence the observation process (Table 4), and these candidate covariates of $q$ and $\sigma$ were evaluated jointly in a preliminary

**Table 4. Candidate covariates for each generalized linear model (GLM) of key parameters in the white pine-Clark's nutcracker analysis (Fig 2): $q$ = per-minute probability of non-detection, $\sigma$ = scale parameter of the half-normal distribution, $\lambda_N$ = expected abundance of nutcrackers, and $\lambda_W$ = expected value of a time-varying white-bark seed proxy (Table 3).**

| Parameter (GLM) | Candidate covariates[a] |
|---|---|
| $q$ (1) | $\boldsymbol{x}_q$ = {day, hour, noise} |
| $\sigma$ (2) | $\boldsymbol{x}_\sigma$ = {observer, noise, dense cover} |
| $\lambda_N$ (3) | $\boldsymbol{x}_N$ = {elevation, aspect, slope, forest, dense cover, PAS, rMST} |
| $\lambda_W$ (4) | $\boldsymbol{x}_W$ = {day, elevation, aspect, slope, PAS, rMST} |

[a]Fixed and/or random effects of year were also considered; see Eqs (3) and (4).

analysis (S2 File), after linking the observation and population process in a hierarchical Bayesian framework (S3 File).

The focus of our population process, $N_{kt}$, was the latent number of nutcrackers at each point-count station in each year, which we modeled as an overdispersed Poisson process with mean $\lambda_N$ determined by covariates as

$$\log(\lambda_{Nkt}) = \beta_{N0} + \boldsymbol{\beta}_{N.}\boldsymbol{T} + \boldsymbol{\beta}_{N.}\boldsymbol{x}_{Nkt} + \varepsilon_{kt} + year_t + transect_k. \tag{3}$$

$T$ in the second term varied among alternative models as $T = year_t$ (H1) or $T = W_t$ (H2). Additional fixed effects $\boldsymbol{x}_N$ suggested by data exploration (Table 4; S1 File) were evaluated in a preliminary analysis (S2 File) described below. Random effects on $\lambda_N$ were normally distributed with mean zero and included an overdispersion term $\varepsilon_{kt}$ with precision $\tau_\varepsilon$, an effect of $year_t$ with precision $\tau_{year}$, and an effect of $transect_k$ with precision $\tau_{tran}$ to account for both spatial autocorrelation and repeated measures.

Candidate covariates of $q$, $\sigma$ and $\lambda_N$ (Table 4) were evaluated in a two-step procedure (S2 File) involving backward stepwise elimination of covariates while monitoring posterior predictive checks to identify a relatively simple but adequate model of the combined observation and population process. Briefly, observation model development began by filling models (1) and (2) with candidate covariates (Table 4), while avoiding high correlation (Pearson's $\rho$ or Kendall's $\tau > 0.5$) among covariates in the same model. We then applied backward stepwise elimination of covariates with no apparent effect on $q$ or $\sigma$ (those with 95% CRIs overlapping zero), while monitoring Bayesian $P$-values for $p_a$ and $p_d$, which report on model fit to data $j$, $b$ and $y$ (Fig 2). Of the models with Bayesian $P$-values between 0.2 and 0.8, the simplest one was used as the basis for evaluating covariates of the population process. Model (3) was then filled with candidate covariates (Table 4) and, setting $T = 0$ to evaluate $\boldsymbol{x}_N$ in absence of $year$ or $W$ effects, the process above was repeated. The resulting 'adequate' model was used as the basis for exploring spatial or temporal dependence of nutcrackers on whitebark.

For spatial dependence, $T = W_k$ in model (3), where $k$ indexes whitebark cover at the point-count station (Table 3). For temporal dependence, $T = W_t$ and we modeled the missing values of each time-varying seed proxy using an appropriate GLM, such as

$$\log(\lambda_{Wkt}) = \beta_{W0} + \beta_W year_t + \boldsymbol{\beta}_W \boldsymbol{x}_{Wkt} + stand_k + plot_k, \tag{4}$$

where $W_{kt} \sim \text{Poisson}(\lambda_{Wkt})$ for counts like $trees$. We assumed a linear trend with $year_t$, covariate effects $\boldsymbol{x}_W$ (Table 4), and normally distributed random effects with mean zero, including an effect of $stand_k$ with precision $\tau_{stand}$ to account for spatial autocorrelation, and an effect of $plot_k$ with precision $\tau_{plot}$ to account for repeated measures. Linking models of $\lambda_W$ and $\lambda_N$ within a Bayesian framework allowed propagation of parameter uncertainty across trophic levels, and appropriate scaling of the spatial process: we estimated $W_t$ as a derived parameter, allowing nutcracker response at each point-count station ($\lambda_{Nkt}$) to be affected by seed resources at the park scale, as suggested by the high vagility of these birds.

The hierarchical mixture models described here present a challenge for current model selection methods [61–64]. Rather than ranking models, we conducted an exploratory analysis with the dual goal of (i) determining whether seed proxies (Table 3) can explain variation in nutcracker density, and (ii) developing a modeling approach suitable for integrating potentially complimentary datasets from long-term monitoring in a multi-trophic system.

## Parameter estimation

To sample and summarize posterior parameter estimates and their 95% credible intervals (CRIs), we used version 4.3.0 of the JAGS programmable platform for MCMC simulation [65],

called remotely from R version 3.5.3 [66]. JAGS code for our spatial and temporal models appears in S3 File. Convergence of parameter estimates was facilitated by coding years 2005–2016 as 1–10 and by standardizing continuous covariates (mean = 0, SD = 1) with the exception of modeled values of *W*. We assumed random effects were normally distributed with mean 0 and precision $\tau$ $(1/\sigma^2)$, while intercepts and coefficients were distributed as $\beta \sim$ Normal $(\mu_\beta,\sigma^2)$, with vague prior distributions on all hyperparameters as $\mu_\beta \sim$ Normal(0,1000) and $\sigma \sim$ Uniform(0,10). Posterior parameter estimates were drawn from 3 Markov chains of 100,000–150,000 samples each, after thinning by 1 in 50 and discarding the first 50,000–100,000 samples.

We assessed the convergence of each parameter estimate using the Gelman-Rubin potential scale reduction parameter, R-hat, which indicates adequate convergence at values of up to 1.2 [67]. Model goodness-of-fit was estimated using posterior predictive checks, summarized as Bayesian *P*-values [68], at multiple points in each model hierarchy. Support for each model covariate was assessed using the 95% CRI for the estimate of its associated coefficient; the covariate was supported when this 95% CRI did not contain zero. Finally, because our model did not explicitly account for autoregressive processes commonly associated with population time series, we inspected the residuals of $N_t$ and $W_t$ for autocorrelation using *acf* in R [69].

## Results

Mountain pine beetle sign was rare, appearing on only 0.4–6.7% of NCCN trees surveyed each year [40] and on <1% of all whitebark and foxtail pines surveyed in SIEN parks [41]. Blister rust infected <1% of whitebark and 0% of the foxtail pines surveyed in SIEN parks [41]. In contrast, blister rust infected a substantial and increasing percentage of whitebark in both NCCN parks during 2004–2016, rising from 18% to 38% in MORA and from 32% to 51% in NOCA [40]. Given these patterns, we focused on the potential for spatial response of nutcrackers to whitebark pine in the SIEN without reference to pests or pathogens, and on the potential for temporal response of nutcrackers to blister rust in the NCCN without reference to mountain pine beetle occurrence.

### Spatial patterns in nutcracker and whitebark metrics

Our simplest adequate model of nutcracker density for both YOSE and SEKI included *elevation* as a covariate of $\lambda_N$ and no covariates of the observation model parameters *q* or $\sigma$ (Table 4). In YOSE, *elevation* and whitebark cover were highly correlated (Pearson's $\rho = 0.50$), so we considered their effects in separate models (Table 5). In SEKI, *elevation* was correlated only moderately with cover of whitebark ($\rho = 0.32$) and foxtail ($\rho = 0.35$), while whitebark and foxtail showed low correlation ($\rho = 0.09$), so we explored three models to illustrate the separate and additive effects of *elevation* and tree cover in that park (Table 5). Together, these five models suggested that average nutcracker densities were similar in YOSE and SEKI during the monitoring period, and were positively related to whitebark cover reported in park vegetation maps. However, in both parks *elevation* had stronger effects (greater coefficients of standardized covariates) than *W*. Both *elevation* and whitebark cover were positive predictors of nutcracker density in YOSE and SEKI, and foxtail cover was also a positive predictor in SEKI, but no interaction between whitebark and foxtail was supported (Table 5). In YOSE, nutcracker density clearly appeared higher in areas of whitebark cover (Fig 3A). In SEKI, nutcracker density appeared highest at mid latitudes in the eastern portion of the park, an area of strong overlap in whitebark and foxtail cover (Fig 4A and 4B). Additive effects of whitebark and foxtail cover were both supported even when *elevation* was included in the same model: 95% CRIs for

**Table 5. Mean and standard deviation of parameter estimates and model diagnostics for spatial models of Clark's nutcracker density in Yosemite National Park (YOSE) and Sequoia and Kings Canyon National Parks (SEKI).**

| | YOSE models | | SEKI models | | |
| --- | --- | --- | --- | --- | --- |
| | *elevation* | $W_k$ | *elevation* | $W_k \times F_k$ | *full* |
| **Abundance** | | | | | |
| *Nutcrackers/ha* | **0.12, 0.01** | **0.12, 0.01** | **0.12, 0.01** | **0.12, 0.01** | **0.12, 0.01** |
| $\beta_{N1}$ (elevation) | **2.72, 0.20** | - | **1.55, 0.21** | - | **1.43, 0.20** |
| $\beta_{N2}$ (whitebark cover) | - | **0.18, 0.05** | - | **0.36, 0.08** | **0.34, 0.08** |
| $\beta_{N3}$ (foxtail cover) | - | - | - | **0.20, 0.07** | **0.14, 0.07** |
| $\beta_{N4}$ (whitebark:foxtail) | - | - | - | *-0.04, 0.03* [a] | *-0.02, 0.03* [a] |
| **Detection** | | | | | |
| $p_a$ | **0.81, 0.03** | **0.81, 0.03** | **0.84, 0.03** | **0.84, 0.03** | **0.84, 0.03** |
| $\sigma_0$ | **68.29, 1.95** | **68.07, 1.94** | **70.52, 2.06** | **70.60, 2.09** | **70.69, 2.03** |
| $p_d$ | **0.30, 0.02** | **0.29, 0.01** | **0.30, 0.01** | **0.30, 0.01** | **0.30, 0.01** |
| **Bayesian *p*-values** | | | | | |
| $p_a$ | **0.49, 0.50** | **0.51, 0.50** | **0.50, 0.50** | **0.49, 0.50** | **0.49, 0.50** |
| $p_d$ | **0.48, 0.50** | **0.45, 0.50** | **0.46, 0.50** | **0.48, 0.50** | **0.48, 0.50** |

[a]Italics indicate 95% credible interval overlapped zero. (All other effects were supported.)

*W* and *F* effects were (0.19, 0.50) and (0.01, 0.28), respectively, in our 'full' model. Note also that the estimated effect size overlapped for (standardized covariates) *W* and *F* in SEKI.

## Temporal study of nutcracker and whitebark dynamics

In MORA, at least two of our proxies for seed production appeared to trend with *year* (Fig 5), and positive effects of *year* on $\lambda_W$ were supported for both: 95% CRIs on trend were (0.039, 0.100) for *rust trees* and (0.066, 0.114) for *crownkill*. In NOCA, only *rust trees* trended with *year* (0.024, 0.063). Fits for these three simple models—without effects other than trend—were adequate (Bayesian *P*-values ranged 0.28 to 0.55), and neither *elevation* effects nor random *year* effects were supported by these data. Note also that signs of infection used to identify *rust trees* included 'browse' marks left by animals attracted to the sugars at presumed infection sites [40, 44]. Our results upheld that presumption: we found cankers in 65% (MORA) to 70% (NOCA) of browsed trees, a significant association (Pearson's Chi-squared test with Yates' continuity correction: $X^2 = 15.418$, df = 1, $p << 0.001$ for MORA; $X^2 = 14.916$, df = 1, $p << 0.001$ for NOCA). Given this information, we proceeded with modeling joint whitebark-nutcracker dynamics.

Our simplest adequate model of nutcracker density in MORA included *dense cover* as a covariate of $\lambda_N$ and no covariates in the observation model. To this model we added (as a covariate of $\lambda_N$) either *year* or one of our four time-varying proxies for whitebark seed production (Table 3). All five models (Table 6) produced similar estimates of nutcracker density which were almost a factor of 10 lower than in the SIEN parks, even though nutcrackers were detected sooner and over greater distances in MORA, based on unanimously higher estimates of $p_a$ and $\sigma$ than in SIEN parks (cf. Tables 5 and 6). Effects of *trees*, *canker trees* and *rust trees* were not supported (95% CRIs overlapped zero) in their respective models, and posterior predictive checks suggested poorer fit for the observed values of those seed proxies than for *crownkill* (cf. $W_t$ values in Table 6). However, there was little to distinguish *year* from *crownkill* models; there was scant difference between their posterior predictive checks, and all covariates were supported within both models (Table 6). Visual comparison of model predictions also

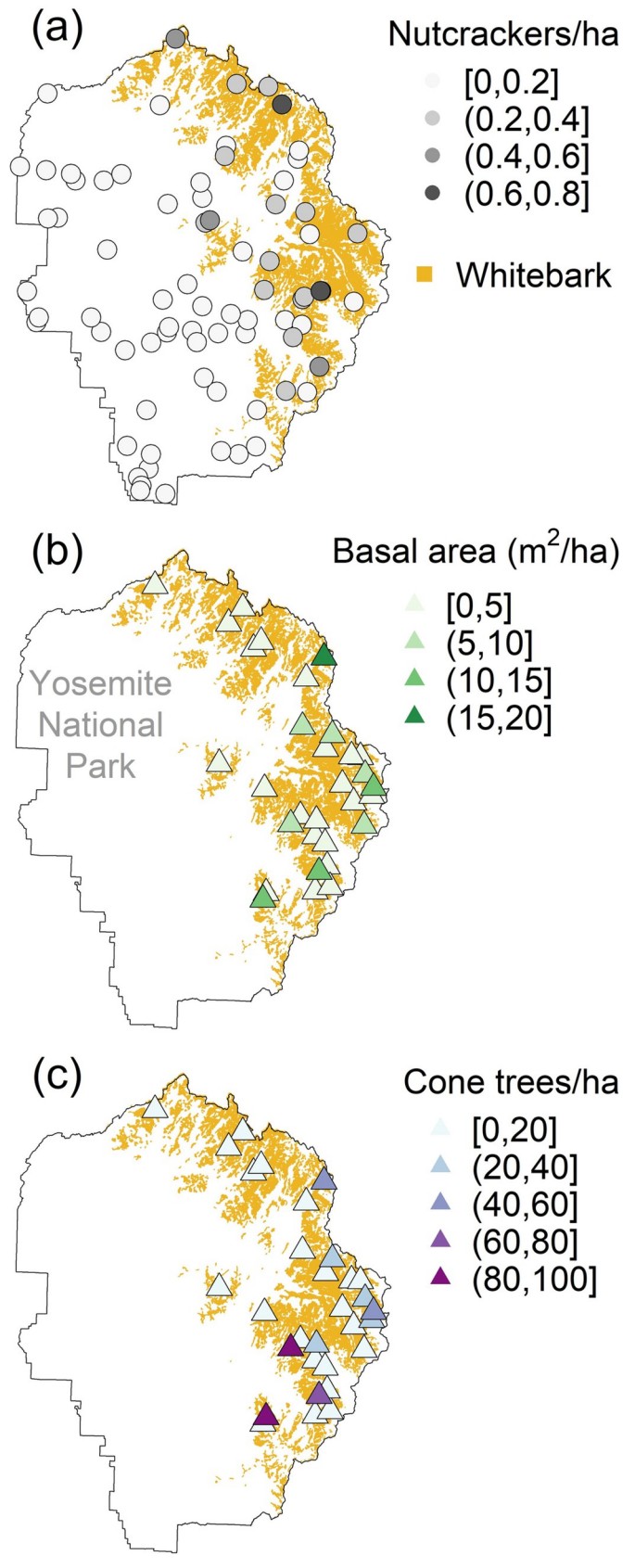

**Fig 3. Modeled density of Clark's nutcracker and raw data on whitebark metrics in Yosemite National Park.**
Estimates of nutcracker density (a), and data on whitebark *basal area* (b) and number of cone-bearing whitebark (c)
were averaged from 2011–2016 surveys. Circles represent the locations of avian point-count transects, and circle
shading represents the relative density of nutcrackers per hectare under the *elevation* model (Table 5). Pearson's
product-moment correlation between *basal area* (b) and *cone trees* (c) was high and significant ($\rho = 0.58$, $p < 0.001$).

suggested strong overlap (Fig 6), although modeling nutcracker response to *crownkill* greatly
reduced the 95% CRI on nutcracker density in MORA during the first half of this study.

In MORA, the temporal trend in nutcracker density was clearly negative (Fig 6), as evi-
denced by negative coefficients on all covariates (Table 6). The 95% CRI for trend with *year*
was (-0.267, -0.105) and for trend with *crownkill* was (-34.76, -11.48), although modeled covar-
iates ($W_t$) were not standardized, so coefficients in our temporal model are not directly compa-
rable. Negative effects of *dense cover* indicate that nutcracker density was higher in more open
habitats. We expected an effect of *dense cover* on detection distance, but none was supported
and model convergence was facilitated instead by allowing *dense cover* to affect $\lambda_N$ directly.
Convergence was also facilitated by restricting our model to high-elevation transects, where 81
of 82 nutcrackers were detected. Thus, we modeled nutcracker abundance across 429 high-ele-
vation point-count stations, out of 1012 total point-count stations in MORA.

Our simplest adequate model of nutcracker density in NOCA included a random *year* effect
as well as *elevation* as covariates of $\lambda_N$, and an *observer* effect on σ. Log(σ) declined with 95%
CRI = (-0.487, -0.172) for observers other than the point-count crew leader. Log($\lambda_N$) rose as
*elevation* increased, with 95% CRI = (1.663, 2.657). Effects of *year* were negative in 2006 with
95% CRI = (-2.462, -0.113) and positive in 2007, 2008 and 2015 with 95% CRI = (0.068, 1.566),
(0.365, 1.723) and (0.646, 1.969), respectively. This model appeared adequate (Bayesian *P*-
value = 0.30 for $p_d$, which is the posterior predictive check incorporating all observed data *j*, *b*
and *y*), and none of the other effects we considered (Table 4) were supported by data from this
park. In NOCA, the variation in nutcracker density from 2005 to 2016 appears to have been
much more dramatic (Fig 6) than can be explained by trends indexed by year or by the appar-
ently monotonic trends in our submodels of NOCA whitebark metrics (Fig 5).

All reported parameter estimates converged according to R-hat and visual inspection of
thinned chains of simulation results. We observed high correlation among several of the
potential predictor variables in our analysis: *day* correlated with *elevation* (Kendall's τ = 0.59),
while *elevation*, presence of *forest* cover and *dense cover* were all somewhat confounded (0.37
< |τ| < 0.77). Only *hour*, *slope* and *aspect* were free of appreciable correlation (|τ| < 0.21). We
found no evidence for residual autocorrelation at any lag in models of nutcracker density for
MORA or NOCA.

## Discussion

We found evidence for strong spatial or temporal variation in Clark's nutcracker detections
within four national parks, and some evidence that the mutualism between nutcrackers and
whitebark pine is currently vulnerable to disruption in the Sierra-Cascades region. In the
Pacific Northwest, where whitebark pines were increasingly infected with white pine blister
rust over a 13-year period, nutcracker detections during our counts were highly variable in
NOCA and declined to zero in MORA. This variation and decline in nutcracker detections
could signal shifting patterns of nutcracker foraging, as discussed below. In the southern Sierra
Nevada, where whitebark stands remain extensive and relatively free of blister rust and moun-
tain pine beetle, our nutcracker detections did not always map onto areas of high whitebark
cover. In SEKI, the time-averaged density of nutcrackers was distributed relatively evenly
across the north-south transition from higher whitebark to higher foxtail cover, and responded

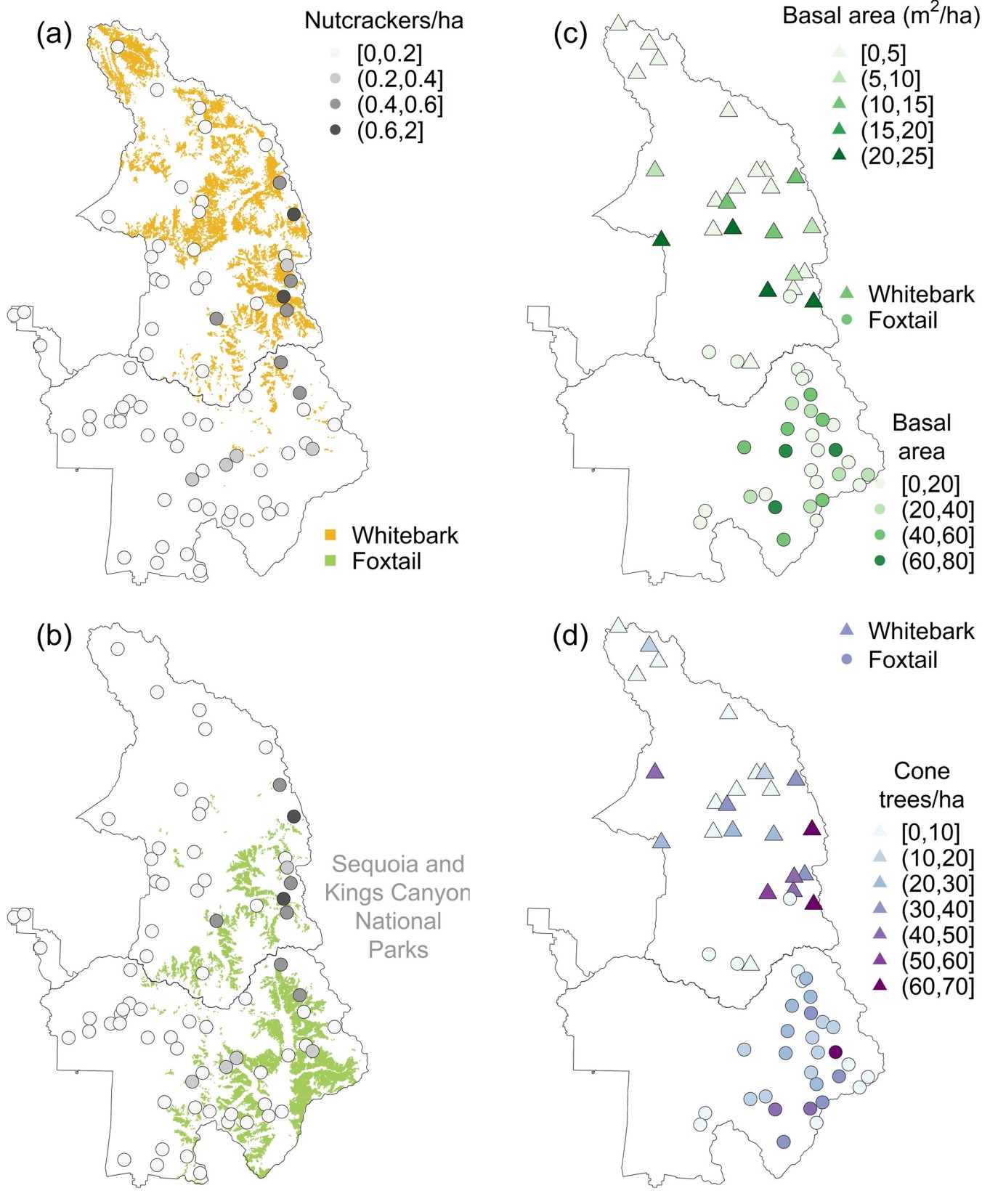

**Fig 4. Modeled density of Clark's nutcracker and raw data on whitebark and foxtail metrics in Sequoia and Kings Canyon National Parks.** Data on nutcracker density (a, b), live tree *basal area* (c) and cone-bearing trees (d) were averaged from 2011–2016 surveys. Circles represent the locations of avian point-count transects, and circle shading represents the relative density of nutcrackers per hectare under the $W_k \times F_k$ model (Table 5), relative to (a) cover of whitebark (gold shading) and (b) cover of foxtail pine (green shading). Symbol shading represents the mean (per survey) of *basal area* (c) and *cone trees* (d) in areas dominated by whitebark (triangles) or foxtail pine (circles). Correlations between *basal area* (c) and *cone trees* (d) were high and significant within each species' sampling frame: Pearson's product-moment correlation was 0.83 for whitebark and 0.94 for foxtail ($p < 0.001$ in each case).

positively to both whitebark and foxtail occurrence. These patterns are concordant with the facultative nature of nutcracker foraging on whitebark—a necessary adaptation given the seasonal and interannual variation in whitebark seed production—and further study appears warranted to determine whether nutcrackers might abandon areas where whitebark cone production is in decline.

Tomback et al. [70] suggested that nutcrackers might disperse foxtail seed, and our results from SEKI appear to add some support for this observation, which is interesting because foxtail seeds are winged for dispersal by wind, and are much smaller (0.027 g each) than whitebark seeds (0.175 g each) [71]. Although individual foxtail seeds would provide less nutrition than whitebark seeds, they might represent an important food resource if—as reported in [41]—there are more cone-bearing foxtail than whitebark, and/or more cones per tree. Whitebark cones begin to mature in August and foxtail cones follow in September [71], but nutcrackers can feed on the immature seeds of both tree species, and might prefer foxtail seeds during certain periods of cone development. If so, the presence of foxtail might attract nutcrackers, which would then be available to cache the maturing seeds of any whitebark within their daily foraging range. It is an open question, however, whether the costs of seed predation will outweigh the benefits of seed dispersal as whitebark populations decline [72]. Spillover of nutcrackers from foxtail into small whitebark populations might perpetuate at least some seed dispersal, but seed predation combined with low seed production might reduce the number of successful dispersal events below a threshold required for effective regeneration.

Nutcrackers readily shift habitats based on resource availability [8, 73], which likely explains why we detected them only after the first week in July, soon after whitebark seeds become edible [48, 74]. It might be surprising, however, that we failed to detect them before this date, or below our high-elevation transects in some parks. Nutcrackers have often been observed foraging and/or nesting in tree species other than whitebark [32, 73–74], including several species found at various elevations in these parks: western white pine (*Pinus monticola*), ponderosa pine (*Pinus ponderosa*), Douglas-fir (*Pseudotsuga menziesii*), Jeffrey pine (*Pinus jeffreyi*), single-leaf pinyon (*Pinus monophylla*), and limber pine (*Pinus flexilis*). They also feed heavily on invertebrates [52], which are certainly widely distributed and available during our breeding-bird surveys. Although nutcrackers fledge by mid June, which is early in our annual survey period, it should be possible to detect them (if present) during foraging activities through late July, when our surveys end. Although the nutcracker detection probabilities estimated from our point-counts were no higher than $p_d \sim 0.30$, the fact that we did not detect nutcrackers outside the elevations and dates during which whitebark and foxtail seeds are edible might suggest that few nutcrackers are using resources in these parks other than the seeds on these trees. In fact, many nutcrackers breed on the east side of the Sierran Crest—which forms the eastern edge of YOSE and SEKI—and cross over to forage in habitats (and parks) on the west side of the Crest only after fledging their young [75], a behavior well-timed for foraging on edible whitebark seeds in SIEN parks. Similar patterns of seasonal migration might explain our observations in NCCN parks. Alternatively, nutcrackers might be difficult to detect when not foraging at high elevations. Nutcracker detection rates have been reported as low and variable during standard point counts, including when radio-tagged birds are known to be present [11,

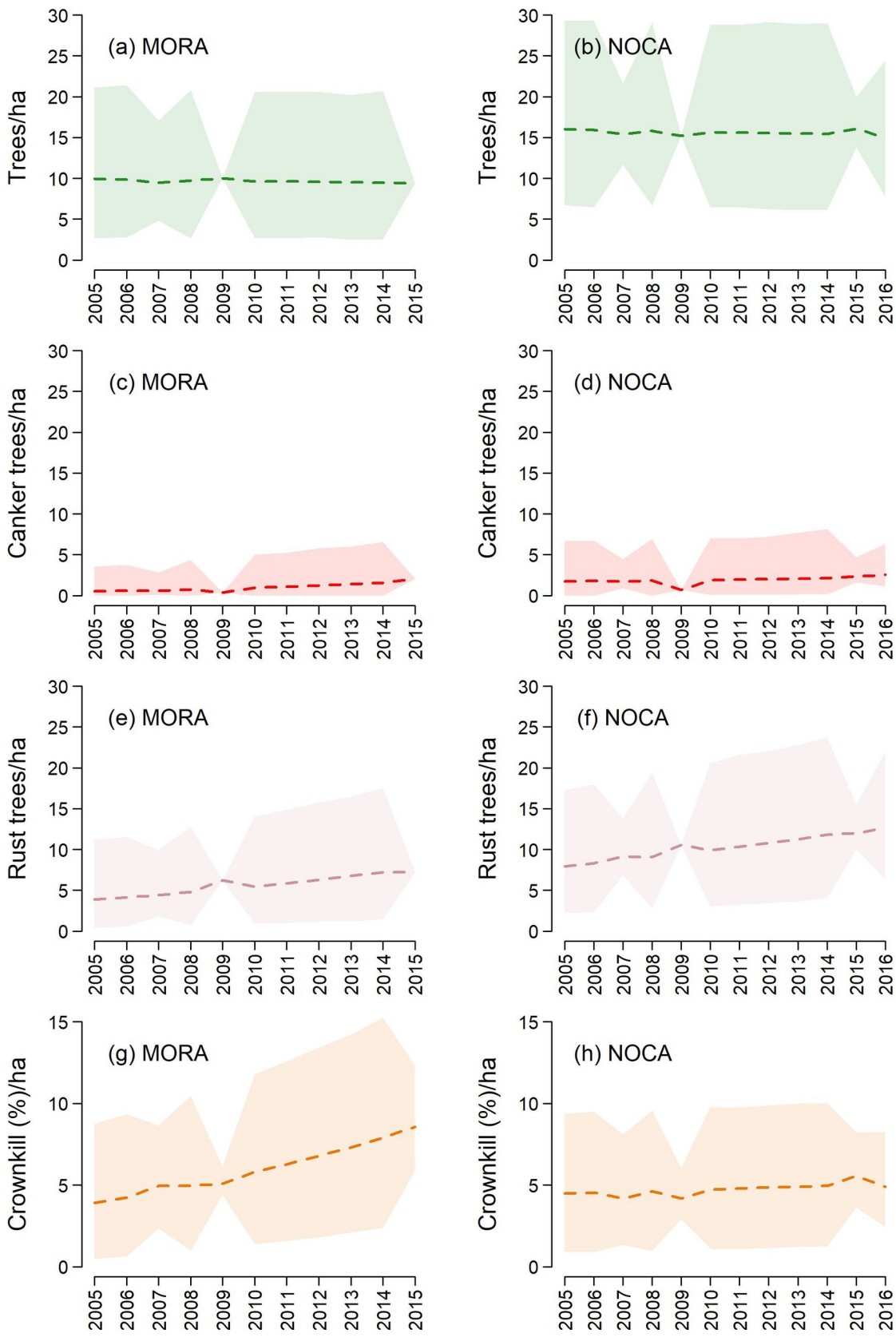

**Fig 5. Annual estimates of whitebark pine metrics in two national parks of the Pacific Northwest.** Means (dashed lines) and 95% credible intervals (polygons) for four potential seed-production proxies (Table 3) based on tree survey data from Mount Rainier National Park (MORA) and North Cascades National Park Service Complex (NOCA). Credible intervals are narrower in years when tree plots were surveyed.

48]. Individual birds also range over very large areas to forage on a wide variety of foods [8, 52, 73], often resulting in very low densities [48].

If nutcrackers are accessing these park habitats mainly to harvest whitebark seeds, then we might expect nutcracker presence to track this seed resource. The temporal decline of nut-crackers in MORA was well supported by our analysis, and could be explained by whitebark crown mortality, which should correlate with seed production [76]. Unfortunately, year also explained the nutcracker trend, and was confounded with crown mortality in our data, in large part because the lack of annual tree surveys in the NCCN required that we model *crown-kill* as a function of *year*. This, and the fact that nutcracker variation in NOCA did not trend with *crownkill* or *year*, further reduces our confidence in *crownkill* (our proxy for seed resources) as a driver of nutcracker dynamics. However, the modeling framework we have developed will be useful for distinguishing drivers of nutcracker dynamics as more data become available from whitebark monitoring. With continued data from the SIEN parks, an analysis of temporal dynamics will be possible for those parts of the southern Sierra.

It is also important to consider that the nutcracker dynamics we observed, including the steady decline in mean and variance of nutcracker density in MORA, might indicate a shift in local range or foraging phenology, rather than a true population decline. For example, nut-crackers in MORA might be shifting their foraging activities into areas outside that park, as described below. Similarly, the irruptions observed in NOCA might represent strong responses to episodic resource availability, such as opportunistic exploitation of whitebark within the park during years when resources outside the park are not as plentiful. Each of these scenarios is possible due to the extreme vagility of these birds [77, 78]. Satellite-tracking of nutcrackers in MORA and NOCA, and surveys conducted at broad scales during the fall harvest season, could be used to address these hypotheses. Nutcrackers forage over large areas, and regularly

**Table 6. Mean and standard deviation of parameter estimates and model diagnostics for temporal models of Clark's nutcracker density in Mount Rainier National Park.** Models are listed by associated hypothesis (H in Fig 2) and distinguishing effect (*T*).

| | MORA models | | | | |
|---|---|---|---|---|---|
| | *year* (H1) | *trees* (H2) | *canker trees* (H2) | *rust trees* (H2) | *crownkill* (H2) |
| **Abundance** | | | | | |
| *Nutcrackers/ha* | 0.014, 0.004 | 0.014, 0.004 | 0.014, 0.010 | 0.014, 0.004 | 0.013, 0.003 |
| $\beta_{N1}$ (*dense cover*) | -0.951, 0.354 | -0.627, 0.331 | -0.651, 0.341 | -0.630, 0.301 | -0.584, 0.353 |
| $\beta_{N2}$ (*T*) | -0.190, 0.042 | -1.748, 1.411[a] | -4.192, 2.238 | -1.374, 0.476 | -22.811, 5.872 |
| $\beta_{W1}$ (*year*) | - | 0.003, 0.024 | 0.121, 0.065 | 0.053, 0.042 | 0.099, 0.013 |
| **Detection** | | | | | |
| $p_a$ | 0.91, 0.05 | 0.92, 0.04 | 0.94, 0.08 | 0.91, 0.05 | 0.92, 0.04 |
| $\sigma_0$ | 108.07, 7.50 | 108.57, 7.65 | 109.51, 9.11 | 108.38, 7.77 | 108.88, 7.87 |
| $p_d$ | 0.25, 0.03 | 0.26, 0.03 | 0.28, 0.05 | 0.25, 0.03 | 0.26, 0.04 |
| **Bayesian *P*-values** | | | | | |
| $p_a$ | 0.50, 0.50 | 0.49, 0.50 | 0.51, 0.50 | 0.49, 0.50 | 0.50, 0.50 |
| $p_d$ | 0.30, 0.46 | 0.28, 0.45 | 0.19, 0.41 | 0.29, 0.45 | 0.28, 0.45 |
| $W_t$ | 0.40, 0.49 | 0.74, 0.44 | 0.19, 0.39 | 0.26, 0.44 | 0.45, 0.50 |

[a]Italics indicate 95% credible interval overlapped zero. (All other effects were supported.)

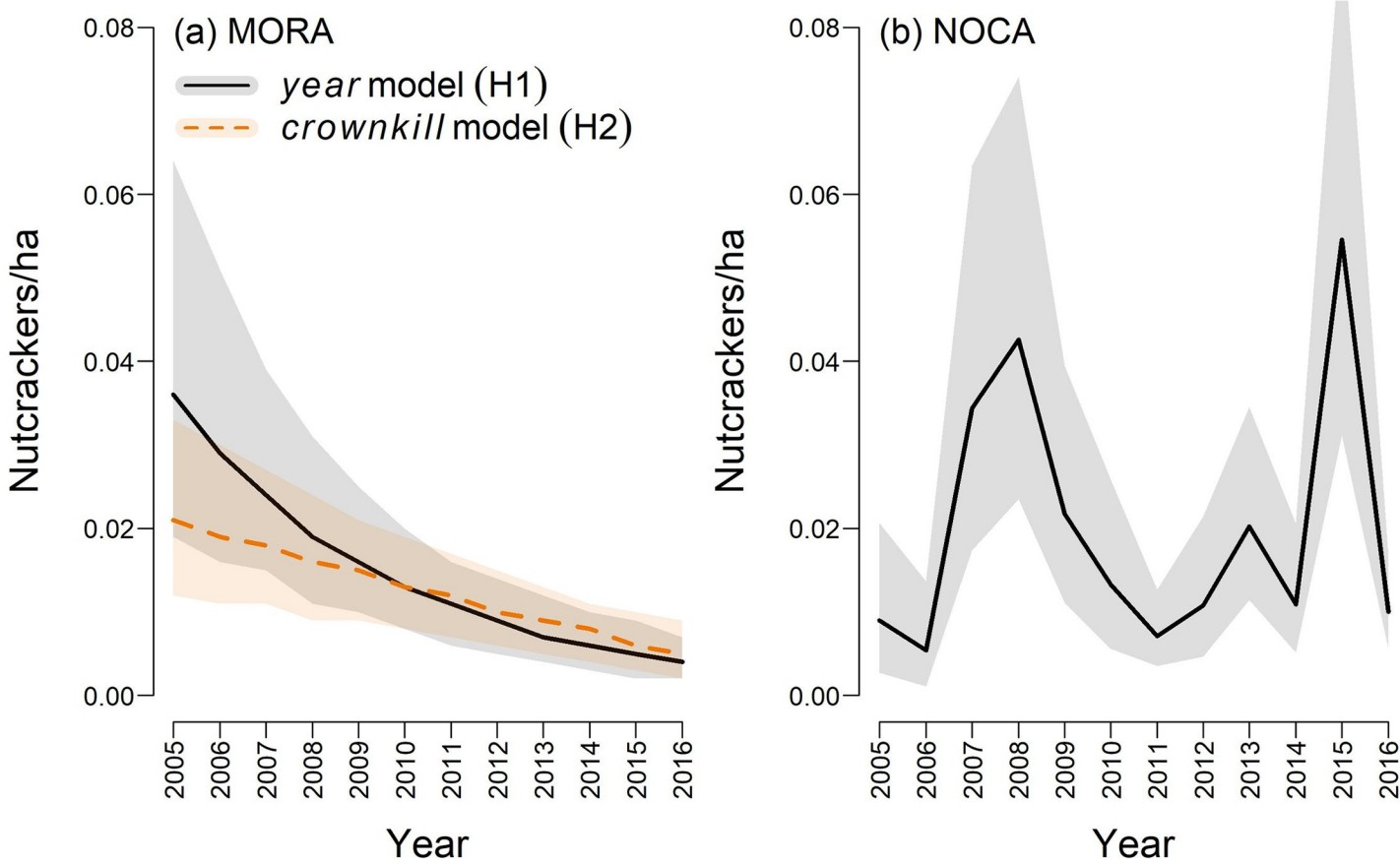

**Fig 6. Modeled estimates of Clark's nutcracker density in Mount Rainier National Park (MORA) and North Cascades National Park Service Complex (NOCA).** Annual estimates were based on models without (solid black curve) or with (dashed orange curve) an effect of whitebark *crownkill* on nutcrackers per hectare. Because most (81/82) nutcracker detections in MORA occurred within high-elevation transects, we did not extrapolate densities to lower elevations in that park. In NOCA, nutcracker density was estimated across all elevational strata, but was unrelated to whitebark trends (Fig 5). Shaded regions represent 95% credible intervals on nutcracker density in each year.

irrupt locally when cone crops are high [77, 78]. Vander Wall et al. [33] documented thousands of nutcrackers moving through the Great Basin over three autumn seasons, and Schaming [48] observed four satellite-tagged nutcrackers moving up to 650 km from the Greater Yellowstone Ecosystem to Utah, where at least three of them overwintered and then returned the following year. Given the nutcracker's clear ability to respond to local resources, the steady decline in mean and variance of nutcracker density in MORA suggests to us that nutcrackers have been shifting their activities outside this park in a deterministic manner.

Several patterns suggest that nutcracker trends might be tied to resources other than the whitebark in these parks. The distribution of nutcracker detections among areas varying in whitebark cover, relative to foxtail cover, provides some support for previous reports of nutcrackers foraging on several pine species [8, 31, 32]. Although nutcrackers prefer whitebark seeds because of their high energy content [9, 10], there must be a threshold below which it is not energy-efficient to seek rare seeds [23]. This potential threshold is currently under study by tracking individual birds (Schaming, personal communication). For example, vegetation maps [79] indicate high whitebark cover just east of both MORA and NOCA, where high cover of whitebark could help explain the low and fluctuating densities of nutcrackers in these parks as a product of spillover. Recent data from seven nutcrackers tracked by satellite just east

of NOCA, shows that these birds move long distances to access varied resources: during a time of year when they were not harvesting mature whitebark pine seeds, the median home range was 31,068 ha (mean = 38,294; range = 17,561–62,974) for these birds (Schaming, unpublished data). Previous radio-tracking in the Cascades demonstrated that nutcrackers ranged 4–29 km from their home range center and covered up to 115 km$^2$ during seed harvesting [11]—and these might be conservative estimates because longer-distance movements can be missed in radio-tracking studies. In other studies, nutcrackers have been observed moving up to 32.6 km between harvest and caching locations [74], and in every season of the year they have been observed moving over 20 km away from their core range (Schaming, unpublished data). This vagility, combined with high cover of whitebark just outside these parks, suggests that nutcracker dynamics within parks will be affected by resources within the larger landscape. Whitebark cone crops tend to be asynchronous at larger scales [80], so that a local decline in cones can be buffered by an increase in cones elsewhere. Even a moderate whitebark cone crop can fail to retain these birds: In the Greater Yellowstone, despite a moderate regional whitebark cone crop in 2015, the majority of tagged birds (71%, n = 5) emigrated and wintered in a diverse variety of habitats up to 650 km away, including limber pine, ponderosa pine, pinyon pine and Douglas-fir, before returning to Wyoming in summer 2016 [48].

Given this potential for long-distance movement and resource sampling, our observation that nutcrackers declined only in MORA and not in NOCA suggests that resources differ between these parks or between the regions surrounding each park. Certainly blister rust has removed more of the whitebark resource in MORA than in NOCA, although trends in NOCA are likely to follow suit [40]. Vegetation maps [79] suggest that whitebark and Douglas-fir constitute the primary sources of nutcracker forage in both parks, aside from invertebrates, and both tree species appear to be more common in NOCA than MORA. In the Greater Yellowstone ecosystem, nutcrackers selected home ranges with disproportionately high cover of Douglas-fir, and foraged heavily on Douglas-fir cones [36, 52]. Douglas-fir/ponderosa pine habitat is also common just east of NOCA, which should help support nutcrackers because they regularly forage on ponderosa pine seeds [11].

In the process of analyzing these data, we identified several challenges relevant to studying this mutualism in this region. First, because nutcrackers make use of several resources, the distribution of these various resources should be considered in study design and monitoring strategies. Although nutcrackers typically prefer whitebark seeds when available [8, 48, 73] this study and others [11] suggest that nutcrackers harvest other foods even when whitebark seeds are available. Cone crops and other resources (e.g., insects) should be assessed both within and, where possible, outside these parks. Second, seasonal variation in resource use requires that we match dates of nutcracker monitoring to the dates of resource use. The timing of standard breeding-bird point-counts might not coincide with the season(s) in which nutcrackers are most frequent or detectable in these parks. Monitoring in the late summer and fall should offer the best opportunity to detect nutcracker use of mature five-needle pine seeds. Third, cone crops should be assessed more directly and more frequently. Years of higher cone production might have been missed in our study because trees were surveyed intermittently. Data are sparse or nonexistent on cone production in whitebark and other tree species used by nutcrackers (at least in this region). Using proxies of potential cone production, such as *crownkill*, could be an effective way to characterize the resource used by nutcrackers. The inverse relationship between *crownkill* and cone production has been noted previously [76], and provides an attractive metric of nutcracker resources. Alternative metrics and alternative models should be considered, however, and long-term monitoring under the NPS Vital Signs program can support this goal. Data from monitoring programs can be used to discriminate among our hypotheses and others in an iterative process of model updating and accumulation of evidence

to advance our understanding and guide resource management [81]. Finally, the spatial scale of monitoring should match the vagility of the seed predator. This might mean augmenting data collected at the scale of tree survey plots with remote sensing of vegetation, and perhaps leveraging recent advances in determining the migratory connectivity of bird populations through isotopic analysis [82], genetics [83] and tracking technology [84].

## Conclusions

The mutualism between Clark's nutcracker and whitebark pine appears vulnerable to disruption, especially where extrinsic stressors threaten these species. The distribution of nutcrackers in national parks of the Pacific Northwest might be changing in response to whitebark mortality caused by blister rust, but the vagility of the nutcracker, the broad distribution of whitebark, and the irruptive dynamics fueled by masting introduce challenges for monitoring, modeling and conserving these resources at relevant scales. The flexible modeling framework we exemplify here is suitable for integrating data from current Vital Signs monitoring efforts targeting nutcrackers and whitebark, and can be adapted to accommodate a variety of changes in the spatial distribution, timing and frequency of monitoring efforts. Monitoring and modeling the dynamics of natural populations across large protected areas can provide insight on the scale of key ecological processes.

## Supporting information

**S1 File. Data exploration.**
(DOCX)

**S2 File. Model development.**
(DOCX)

**S3 File. Spatial and temporal model scripts.**
(DOCX)

## Acknowledgments

We thank numerous field crews trained and supervised by the National Park Service (NPS) and The Institute for Bird Populations (IBP) for collecting tree and bird data. Special thanks to Mignonne Bivin and Laurie Kurth for leading survey crews, Natalya Antonova for GIS support, and Lise Grace for support with data entry and quality assurance. Project assistance was also provided by staff from the NPS Inventory and Monitoring (I&M) Division, North Coast and Cascades I&M Network, Sierra Nevada I&M Network, Mount Rainier National Park, North Cascades National Park Service Complex, Olympic National Park, Sequoia and Kings Canyon National Parks, and Yosemite National Park. This is Contribution Number 679 of The Institute for Bird Populations. Any use of trade names is for descriptive purposes and does not constitute endorsement by the U.S. government.

## Author Contributions

**Conceptualization:** Chris Ray, Regina M. Rochefort, Jason I. Ransom, Jonathan C. B. Nesmith, Sylvia A. Haultain, John R. Boetsch, Rodney B. Siegel.

**Data curation:** Chris Ray, Regina M. Rochefort, Jonathan C. B. Nesmith, John R. Boetsch, Mandy L. Holmgren, Robert L. Wilkerson.

**Formal analysis:** Chris Ray.

**Funding acquisition:** Chris Ray, Regina M. Rochefort, Jason I. Ransom, Sylvia A. Haultain, Rodney B. Siegel.

**Investigation:** Chris Ray, Regina M. Rochefort, Jonathan C. B. Nesmith, Sylvia A. Haultain, Taza D. Schaming, Mandy L. Holmgren, Robert L. Wilkerson, Rodney B. Siegel.

**Methodology:** Chris Ray, Regina M. Rochefort, Jonathan C. B. Nesmith, Sylvia A. Haultain, John R. Boetsch, Mandy L. Holmgren, Robert L. Wilkerson, Rodney B. Siegel.

**Project administration:** Regina M. Rochefort, Jason I. Ransom, Rodney B. Siegel.

**Resources:** Regina M. Rochefort, Jonathan C. B. Nesmith, Taza D. Schaming, John R. Boetsch, Robert L. Wilkerson, Rodney B. Siegel.

**Software:** John R. Boetsch.

**Supervision:** Regina M. Rochefort, Jason I. Ransom, Rodney B. Siegel.

**Validation:** Chris Ray, Taza D. Schaming, John R. Boetsch, Mandy L. Holmgren, Robert L. Wilkerson, Rodney B. Siegel.

**Visualization:** Chris Ray.

**Writing – original draft:** Chris Ray.

**Writing – review & editing:** Chris Ray, Regina M. Rochefort, Jason I. Ransom, Jonathan C. B. Nesmith, Sylvia A. Haultain, Taza D. Schaming, John R. Boetsch, Mandy L. Holmgren, Robert L. Wilkerson, Rodney B. Siegel.

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
