## [Decision Letter · Decision Letter 0]

28 Jan 2020

PONE-D-19-34230

Assessing trends and vulnerabilities in the mutualism between whitebark pine (*Pinus albicaulis*) and Clark’s nutcracker (*Nucifraga columbiana*) in national parks of the Sierra-Cascade region

PLOS ONE

Dear Dr Ray

Thank you for submitting your manuscript to PLOS ONE. After careful consideration, we feel that it has merit but does not fully meet PLOS ONE’s publication criteria as it currently stands. Therefore, we invite you to submit a revised version of the manuscript that addresses the points raised during the review process.

In general, I agree with both referees, who think that the contribution of this study could be valuable. Whitebark pines are a key species in the area, but indirect effects of different stressors can decouple their obligate mutualism with nutcrackers. However, I have four major concerns. Firstly, as pointed out by the second referee, methodology was quite difficult to follow. I had the impression that simpler and more adequate models could be applied.  Secondly, spatial relationships between nutcrackers and pines could not be formally tested. However, there are strong interpretations about these results in the abstract, discussion and conclusions. I strongly suggest, either to find a way to formally test these spatial correlations or to remove these results and focus on NCCN dataset. Referee 2 provides interesting ideas in this regard. Thirdly, I agree with this referee that applying a gam or including year as a continuous variable may not be appropriate since only three year data was available. Finally, methods of model selection were not clear or correct (see referee 2) and probably a more robust biological model could help clarifying the results (see below).

How nutcracker abundance was modeled was a little bit confusing to me. If the data available are point counts and the authors want to estimate abundances. Why not applying the Royle and Nichols model (2003)? Since individuals are not marked, your estimates of abundances can be biased due to double-counting. There are also other interesting options, like corrected Poisson models (see p. 305 from “Applied hierarchical modeling in ecology” book). If none of these options are applied please refer to bird activity rather than abundance. In this context, I also find interesting the comments made by referee 1 about talking about co-ocurrence rather than mutualism.  I also found the observation model too complex.  I suggest collapsing availability and perception on the probability of individual detection. This would simplify the model without losing information about your target parameters. Finally, if temporal trends are being tested; why not including previous year counts in the nutcracker model? If it is not possible, why not including a spatial autocorrelation structure in the residuals rather than year as a fixed effect?

If I understood correctly, different sets of models were tested. Since there were many covariates, I was wondering whether there could have been type I errors.  My suggestion is to test one theoretical model, which includes only the covariates that are important for the state and observation process. This would also simplify the methodology. If multiple tests are performed, please control for type I errors. Maybe, type I errors are causing the contradictory results of a strong overlap between the crown-kill and the null model (fig. 6) and the strong effect of this covariate.  In addition, if multple models are tested model selection methods need to be clarified or corrected (see referee 2). Although I like the idea of figure 2 and I am aware that the information is in the text, the model of temporal trends was very difficult to follow. Writing down the formulation of the hierarchical model and including covariate names on table 3 would make the reading more straightforward.

Regarding gam models, I think they may not be correct since, if I understood correctly, for vegetation surveys only data from 2004, 2009, 2015 are available. If so, these models may not provide trustable information for vegetation surveys since only three points are available.  As pointed out by referee 2, I suggest authors to remove these analyses and focus on the analyses of fig. 2. In addition, since covariates of white bark pines were predicted using data of these three surveys. It would be interesting to perform posterior predictive checks of the model, just to ensure that increased uncertainity throughout the model dit not lead to unreliable results

Regarding results, there are many figures some of them with similar information that could be reduced (i.e. fig. 7).  I also agree with both referees that including a summary table of the results will make the manuscript more attractive.

Finally, the description of the surveys and the study sites was difficult to follow because too much information was provided (see minor comments below) and most importantly, in some parts it was mixed with the “experimental design”. For example, until data-analysis section I was wondering if the plots of vegetation and transects of avian surveys were in the same area.  I still have this doubt for the NCCN dataset.  In addition, I am not sure that 7 minute surveys can be used as different visits. Aren’t they highly correlated? In sum, please separate well the description of the study area and the surveys. Clarify if vegetation plots and avian surveys were located in the same area for NCCN dataset. If not, please clarify how can the two processes be linked and modeled simultaneously.

Overall I liked the idea of the work. It is clear that the authors have vast knowledge of the pine-nutcracker mutualism and that they are able to apply complex statistical analyses. Nonetheless, I had some doubts about model specifications, which made me wonder if the results were only reflecting type I errors, and if the dataset was adequate for their aims. If these issues are clarified or solved, I believe the work can make important contributions for our understanding of the nutcracker-white pine mutualism and its conservation.

We would appreciate receiving your revised manuscript by Mar 13 2020 11:59PM. To enhance the reproducibility of your results, we recommend that if applicable you deposit your laboratory protocols in protocols.io, where a protocol can be assigned its own identifier (DOI) such that it can be cited independently in the future. For instructions see: http://journals.plos.org/plosone/s/submission-guidelines#loc-laboratory-protocols

We look forward to receiving your revised manuscript.

Kind regards,

Teresa Morán-López

Academic Editor

PLOS ONE

Additional Editor Comments:

Minor comments

L36-39. My suggestion is to use less technical words in the abstract.

L56-58 An example with plant-animal mutualisms would be better.

L102-109 Please summarize

Fig. 2  I have some doubts about how the processes are connected.  For instance the probability that a nutcracker is available for detection does not influence the predicted number of nutcrackers (n) but the observed data (y).

L123-126 I suggest to finish the introduction summarizing the aims of the studies and how they were approached, rather than with a technical description of the model.

L137-138 Too much detail.

Fig. 1 I would add an explanatory figure of the sampling design. As another pannel of this one.

Table 1-2 I would move them to a supplementary material.

L180 I guess panel is the altitude but it was confusing to me.

L182-183 Please clarify

L191 Please state that these intervals are used as “visits” for imperfect detection.  Justify why potentially autocorrelation effects can be disregarded.

L194 How was ambient noise measured?

L196 How was aspect measured?

L207-208 At the end, it was not clear to me which variables were used. Please clarify and summarize your climatic variables.

L213-216 This is too vague. What is the particular suggestion of  Bivand?

L221-229 I thought gams were used for selecting covariates, but then they are presented as results. Nonetheless, I am not sure that they are correct.

L244 Distance sampling was not included in the description of the survey.

Table 3. Why the probability of perception is modeled with a log-link instead of a logit link?

L287 First time that canker is stated, please specify this in the survey, and describe it for non-specialized readers.

L289 Why the proportion of trees with cones was not included?

L316 Please specify the value of the precision parameter and the value of the priors.

L340-341 Please move this information to the data description. I was wondering this for a long time.

L349 None of these analyses support the discussion and conclusions of the spatial relationships between pines and birds.

Table 4 Please move to an appendix

L388. Only three years, then I would not use year as a continuos covariate but as a factor.

L437 These estimates seem too big for a standardized covariate.

2. In your Methods section, please provide additional location information of the study area, including geographic coordinates for the data set if available.

3. In your Methods section, please provide additional information regarding the permits you obtained for the work.

Please ensure you have included the full name of the authority that approved the field site access and, if no permits were required, a brief statement explaining why.

Reviewers' comments:

Reviewer's Responses to Questions

**Comments to the Author**

1. Is the manuscript technically sound, and do the data support the conclusions?

Reviewer #1: Yes

Reviewer #2: Yes

2. Has the statistical analysis been performed appropriately and rigorously? 

Reviewer #1: Yes

Reviewer #2: No

3. Have the authors made all data underlying the findings in their manuscript fully available?

Reviewer #1: Yes

Reviewer #2: No

4. Is the manuscript presented in an intelligible fashion and written in standard English?

Reviewer #1: Yes

Reviewer #2: Yes

5. Review Comments to the Author

Reviewer #1: Ray et al. use data from four national parks in western North America to investigate the co-occurrence of whitebark pine (PIAL) and Clark’s nutcrackers (CLNU), as well as its temporal trends as the trees are declining due to blister rust. The data from the southern parks (SEKI; YOSE) shows that CLNU detections are more frequently detected in areas with pines, whether just PIAL (YOSE) or also foxtail pine (SEKI) The extensive analysis of time series from the two northern parks reveals that CLNU sightings are becoming less frequent in the Mount Rainier area (MORA), while they fluctuate strongly in the northern Cascades (NOCA). Based on the results, the authors then discuss that CLNU are flexible in their association with PIAL, as expected in this facultative mutualism, and that PIAL declines in the park may not affect the population sizes per se, but may force the birds to forage for their preferred seeds in other areas, outside of the managed NPS boundaries.

The paper is well written, addresses an important biological and conservation issue, and refrains from broad interpretations beyond the investigated phenomenon. Aside from some minor comments on the discussion, I have three suggestions aimed to improve the clarity of the article for the readers.

For full disclosure, I am not familiar with Bayesian modeling and could thus not assess whether the details of the analysis were performed correctly. The selection of variables, as well as the sampling regimens and data sources are satisfactory, however, and the authors have extensive experience and are well-regarded for these types of models. Therefore, I have no reason to doubt any of the methods or results presented.

- While co-occurrence, by definition, underlies the mutualism between the two species, it would be helpful to be clear about the fact that here, only co-occurrence is considered. Rather than speaking of the decline in the mutualism, it may be more accurate to speak of the “co-occurrence underlying the mutualism”, or “… that underlies the mutualism”, or something along those lines. This is just to point out that the actual movement ecology of foraging, seed caching, recovery, etc. is also involved in the mutualistic interaction, none of which is addressed by data in this MS.

- A table that summarizes the findings from the complex model may be helpful. While most the findings are clearly communicated in the text and the figures, a simple table may help convey the wide array of variables considered and allows the authors to highlight those that were found to be important.

- A minor wording issue, but “crown kill” makes a good proxy for potential seed production, but realized seed production can vary tremendously between years and cannot be captured with this metric.

- A stylistic issue which is really just a matter of preference: perhaps, the authors could consider flipping the color scheme in Figs. 3-7, so that the tree data is in grey-scale and the bird data contains color. Since the paper is more focused on the effects of tree demise on bird density, this would be better reflected and make it easier to interpret some of the more cluttered figures.

- Finally, consider including the outline of the NCCN and SIEN in Fig. 1

Details:

- L 36: “trophic dependence” is not really what the models are investigating, it’s really about co-occurrence dynamics. No foraging data was collected.

- L 102: “Vital Signs”. Is this a NPS program? If so, consider describing it in short. The sudden capitalization is confusing.

- L156: “4421 m in elevation”

- L164: “3,091 m in elevation”

- Fig. 4a and b; confusing legend panel in Fig., as the tree species appears to be listed under CLNU as “detected”, “not detected” and “whitebark (or foxtail) pine” [consider removing last indentation]

- Fig. 5: why are the CIs so small for 2009? Make sure to mention absence of CLNU in 2016 in MORA in caption

- L 491: the data from SIEN almost seems like an afterthought in this paper. Have you considered showing the 6-year time series without modeling it? Otherwise, the paper may even benefit from dropping the SIEN part, since it does not tell us anything about temporal trends…

- L498-515: the two paragraphs are a bit contradictory, as you state in the first one that foxtail may attract CLNU which may then depredate PIAL seeds, and in the second that attracting to CLNY with foxtail may increase seed dispersal. This may be reconciled with some word-smithing.

- L553pp. Consider what may happen to the variability, not just the mean of nutcracker attendance. Perhaps they may just show up less frequently, or less reliably

- L592: here you state 650 miles, earlier they were kilometers with the same reference. Please reconcile

- L 620: “proxies of potential cone production”

Congratulations to the authors on a well-done and important study.

(PS - I assume the data will be made publicly available)

Reviewer #2: Plos one review: Trends in the whitebark pine-Clark’s nutcracker mutualism

Thank you for the opportunity to review this manuscript. It was generally comprehensive and well-written with relevant conclusions to warrant publication in Plos one. I appreciate that the authors attempted to examine multiple facets of potential mutualism and association between nutcracker abundance and whitepine distribution, productivity, and pathogens. The integrated analysis examining the effects of whitepine trends on nutcracker abundance was particularly creative. My one general comment is that the paper is trying to do so much, it is quite difficult to follow through the methods and results. The authors are dealing with two primary analyses (temporal trend and spatial pattern) at 4 different sites, and each analysis has several different response variables and covariate sets. Instead of Figure 2, which provides a schematic of the N-mixture integrated model structure, perhaps a schematic of how each analysis or summary is structured would be much more helpful. Or a table of candidate models for each analysis. Something that structures that information and puts a reference in one place. Also, I suggest using additional sub-headings for different analyses or sites or something that helps readers index what information belongs to what. Other than restructuring the Methods/Results sections to provide additional clarity, the authors could include additional language describing model/covariate selection procedures. Otherwise, my comments are mostly editorial in nature (see below).

Line item comments:

Line 30-31: Remove scientific name from species because its in the abstract?

Line 64: Are names of species originators required as part of Plos one formatting? Suggest omitting here and elsewhere.

Line 90-91: Suggest revising sentence to something more concise – for example, “Whitebark requires Clark’s nutcrackers to disperse its wingless seeds.

Line 111: This objective seems to incorporate only part of your analyses. The following paragraph (lines 117-126) delves more into methods, but doesn’t succinctly state the objective of each analysis – perhaps reword to focus more broadly on study objectives and reserve methodological discussion to the Methods or Discussion sections.

Line 176: Subheading for avian data?

Line 178: Replace ‘to’ with ‘in’?

Line 194-195: I’m not clear on what ‘cover density’ represents. I suggest briefly defining here and perhaps changing lower and higher cover to lower and higher density in parentheses?

Line 211-220: Your spatial analyses and resulting figures seem a bit clunky/rudimentary. I’m confused why you reduced avian count data to apparent presence/absence despite having counts and information to estimate detection probability. If you used those data, you could create a nutcracker density map across SIEN sites that included whitebark or foxtail DBH, # of cone-producing trees, elevation, etc. as covariates. Summarized in grid cells that approximated the size of the area surveyed per point count. From this information you could provide helpful summary statistics such as XX% of the nutcracker population in SIEN occurred in whitebark dominated habitat, while XX% occurred in foxtail habitat. Or, cells with trees > XX DBH had XX% greater density of nutcrackers. Something more tangible than multiple pie charts over landcover layers that are difficult to interpret.

Lines 221-229: Did these avian exploratory analyses actually feed into any other models? It doesn’t appear so – I would delete this section and the corresponding section in the Results – find a way to incorporate the necessary covariates into a single temporal analysis.

Line 231: Can you be more specific than whitebark dynamics? It would add clarity to define the response variables in the GLMs that fed into the N-mixture model as covariates.

Table 3: Not very informative – W = whitebark metrics and x = other covariates. See general comment, but some detailed summary of what actually was examined in each model component would be useful here and for other analyses.

Lines 297-300: I’m confused as to how you actually conducted model selection. Here you state you included each covariate in a set of candidate models with 0-4 additional covariates. Then on lines 323-325 you state you examined GOF statistics and overlapping 95% CIs, but if you included a varying number of covariates per model, you must have somehow decided which covariate combination was the ‘winner’ in order to interpret coefficients? I’m not sure if this was poorly done or I’m missing the authors’ intent. In the Results section, Bayes-P and LOF ratio are reported as if they provide evidence in support of a specific covariate set – which is an inappropriate use of those metrics. Regardless, if you have a candidate model set for each analysis, it would be worth including it as a table or appendix. Further, if you’re looking for a more formal way to conduct Bayesian model selection, there are few options including WAIC or LOO, which can be implemented pretty easily in R using the LOO package by exporting the log likelihood estimates from your jags model and running a couple lines of code. From that you can obtain an information criterion metric that could be used to formally compare models. Regardless, this section needs to be rewritten to facilitate understanding of the actual model/covariate selection process.

Line 316: Why mean = 1.0 here and not 0.0, which is more typical for uninformative random effect priors?

Line 385-388: So you’re really basing your temporal trends on three data points for whitebark metrics. Is the implication of such a minimal sample size on your ability to make inference warranted in the Discussion?

Line 456-473: So, these results are from a series of ad hoc GAMs on the apparent presence/absence of nutcrackers? Why abandon the results of your N-mixture model to delve into covariate effects from this analysis? Could these covariates not have entered into your N-mixture model, and if they weren’t supported, shouldn’t be discussed further?

Line 482: The Discussion is quite long and focuses extensively on scenarios that call their study into question such as dynamics outside of park boundaries and evidence suggesting nutcrackers aren’t that dependent upon whitebark. I think these sections are important to include but could be shortened significantly. Focus on the interesting results from your study, how they fit in the broader literature, what issues occurred with the data and your ability to make inference, and what could be done going forward.

Line 624: All other citations are numbers, why author and year here?

6. PLOS authors have the option to publish the peer review history of their article (what does this mean?). If published, this will include your full peer review and any attached files.

Reviewer #1: Yes: Mario Pesendorfer

Reviewer #2: No

---

## [Author Response · Author response to Decision Letter 0]

2 May 2020

PONE-D-19-34230

Assessing trends and vulnerabilities in the mutualism between whitebark pine (Pinus albicaulis) and Clark’s nutcracker (Nucifraga columbiana) in national parks of the Sierra-Cascade region

PLOS ONE

Response to review

We appreciate the very careful reviews we received on this ms, and have responded to each comment and concern below. Please see “Response“, throughout.

Dear Dr Ray

Thank you for submitting your manuscript to PLOS ONE. After careful consideration, we feel that it has merit but does not fully meet PLOS ONE’s publication criteria as it currently stands. Therefore, we invite you to submit a revised version of the manuscript that addresses the points raised during the review process.

Response: We have revised our ms in response to each point.

In general, I agree with both referees, who think that the contribution of this study could be valuable. Whitebark pines are a key species in the area, but indirect effects of different stressors can decouple their obligate mutualism with nutcrackers. However, I have four major concerns. Firstly, as pointed out by the second referee, methodology was quite difficult to follow. I had the impression that simpler and more adequate models could be applied. 

Response: We have revised and expanded our explanation of the models, and now provide additional material in tables to help guide the reader through our analyses.

Secondly, spatial relationships between nutcrackers and pines could not be formally tested. However, there are strong interpretations about these results in the abstract, discussion and conclusions. I strongly suggest, either to find a way to formally test these spatial correlations or to remove these results and focus on NCCN dataset. Referee 2 provides interesting ideas in this regard. 

Response: We have removed any text suggesting strong interpretations based on our spatial exploration of the data, and we address the suggestions of Referee 2 below by adding new hierarchical models of the spatial distribution of nutcracker density as a function of whitebark and/or foxtail cover as depicted on park vegetation maps. Although we cannot relate nutcracker density to tree metrics derived from our tree plots, using vegetation map data on tree cover allows us to more formally evaluate the spatial relationships that were suggested by the maps in our original ms.

Thirdly, I agree with this referee that applying a gam or including year as a continuous variable may not be appropriate since only three year data was available. 

Response: We regret any confusion on this point, and we have revised the text and relocated these plots to an appendix more suitable for exploratory data visualization. GAMs were used only as a time-varying analog of box plots, to help the reader visualize any time-varying relationship between nutcracker detection and conditions at our bird point-count stations, including static conditions (e.g., elevation) and variable conditions (climate), and only for conditions that were known for every year (2005-2016). We did not use GAMs to relate nutcracker detection to the temporally sparse data on whitebark metrics. Finally, although we mentioned that “most” tree surveys in NCCN parks occurred in 2004, 2009 and 2015, we did not mean to imply that trees were surveyed in only 3 years. Table 1 shows that trees were surveyed in 4 years in MORA and 5 years in NOCA, and our temporal model was informed by data from all survey years, even years with an incomplete survey (e.g., 2007 in MORA). We have clarified these points by making changes in the text and organization of Methods and Results, and we have moved all GAMs to the supplement. 

Finally, methods of model selection were not clear or correct (see referee 2) and probably a more robust biological model could help clarifying the results (see below).

Response: We have made major revisions to address these issues. Our focal analysis originally involved the comparison of two models (termed here A and B), one without an effect of whitebark on nutcrackers (A) and one with an effect of whitebark (B). In our temporal analysis, these two models produced very similar predictions, so we chose the conservative approach of presenting these predictions graphically, to reveal their overlap, rather than reporting results of model selection. Given the exploratory nature of our analysis and the sparse temporal data on whitebark metrics, we did not want to suggest this was a confirmatory (model-selection) analysis. In our revision, we emphasize this point and also clarify that popular methods of model selection often are inadequate for hierarchical mixed models with overdispersion, such as those we consider (as reviewed in references newly added to our ms: Hooten and Hobbs 2015, Zipkin and Saunders 2018, Plard et al. 2019, Williams et al. 2019). As before, we refrain from drawing conclusions about nutcracker dependence on whitebark based on this exploratory analysis, but we add a discussion of the potential for accumulating evidence for alternative models through long-term monitoring and the evolving information-state approach suggested by Nichols et al. (Nichols JD, Kendall WL, Boomer GS, 2019. Accumulating evidence in ecology: once is not enough. Ecology and Evolution 9:13991-14004. DOI:10.1002/ece3.5836). This discussion brings our ms full-circle to suggest how long-term monitoring of national park resources can be used to guide conservation and answer fundamental questions in ecology.

How nutcracker abundance was modeled was a little bit confusing to me. If the data available are point counts and the authors want to estimate abundances. Why not applying the Royle and Nichols model (2003)? Since individuals are not marked, your estimates of abundances can be biased due to double-counting. There are also other interesting options, like corrected Poisson models (see p. 305 from “Applied hierarchical modeling in ecology” book). If none of these options are applied please refer to bird activity rather than abundance. In this context, I also find interesting the comments made by referee 1 about talking about co-ocurrence rather than mutualism. I also found the observation model too complex. I suggest collapsing availability and perception on the probability of individual detection. This would simplify the model without losing information about your target parameters. 

Response: We appreciate your astute suggestions about the difficulty of estimating density from data on untagged individuals. We are familiar with the extensive literature on modeling density from bird point-count data, and we have selected from that literature to build a model of nutcracker density that is particularly appropriate for our data. The core of our model derives from a model first published by Amundson, Royle and Handel (2014), which was derived from models put forth in Royle and Nichols (2003) and other publications from that research group. Specifically, it incorporates data on distance to detection and time to detection that together allow more precise and less biased estimates of density than methods that do not parameterize detection probability in terms of perceptibility by distance and availability by time. However, we agree that the complexity of this well-published model perhaps overshadowed the novel contribution of our ms, which was to link the model of nutcracker density with a model of whitebark dynamics. To better emphasize the novel contribution of our ms, and to reduce the complexity of our Methods section, we have reduced our discussion of nutcracker detection specifics (like availability and perceptibility) by making more frequent reference to previous publications of this model. We hope this addresses your suggestion that we simplify the ms, without compromising our description of key parameters (like overall detection probability). 

Finally, if temporal trends are being tested; why not including previous year counts in the nutcracker model? If it is not possible, why not including a spatial autocorrelation structure in the residuals rather than year as a fixed effect?

Response: We considered autoregressive forms of our temporal model, but we were not able to achieve convergence for models that included an autoregressive term, possibly because the decline in nutcrackers was so monotonic that there was high covariance between year t and t-1 terms. Another consideration is that the literature on Clark’s nutcracker contains many examples of irruptive dynamics, meaning we might not expect strong temporal autocorrelation in the dynamics of this species. We now address this issue in the revised ms by reporting results of our examination of model residuals for autocorrelation. Regarding your question about spatial autocorrelation, we accounted for that by including a random effect of transect. 

If I understood correctly, different sets of models were tested. Since there were many covariates, I was wondering whether there could have been type I errors. My suggestion is to test one theoretical model, which includes only the covariates that are important for the state and observation process. This would also simplify the methodology. If multiple tests are performed, please control for type I errors. Maybe, type I errors are causing the contradictory results of a strong overlap between the crown-kill and the null model (fig. 6) and the strong effect of this covariate. In addition, if multple models are tested model selection methods need to be clarified or corrected (see referee 2). 

Response: The problem of Type I error is interesting in this case. On one hand, we’re working within a Bayesian framework that supports parameter estimation rather than hypothesis testing, and we had not characterized any of our models as a null hypothesis. Instead, we focused on whether there was a) support for the dependence of nutcrackers on whitebark dynamics in our joint whitebark-nutcracker model, and b) clear impacts of that dependence on predictions of nutcracker density. We simply estimated the effect of the whitebark term in our model of nutcracker density, and assumed the effect was supported if its 95% credible interval did not overlap zero. Although we found a) support for an effect of whitebark on nutcracker dynamics, we also found that b) nutcracker predictions overlapped substantially between models with and without this effect. Given these results, we decided that the evidence for nutcracker dependence on whitebark was inconclusive but warrants further investigation. On the other hand, we also considered four possible effects of whitebark on nutcracker dynamics (1 = % crownkill, 2 = # rust trees, 3 = # canker trees, 4 = # live trees), which could increase our potential for finding at least one effect that was supported. We note, however, that the direction of each effect was as expected (negative for effects 1-3 and positive for effect 4), which suggests an underlying process other than random chance. Finally, because there was high covariance among these four whitebark metrics, these were not actually four independent tests. We have revised text in Methods, Results and Discussion sections to clarify that this is an exploratory rather than hypothesis-testing exercise, designed to advance the integration of long-term monitoring datasets, but still limited by the early stages of these programs.

Our approach is similar to your suggestion of testing “one theoretical model, which includes only the covariates that are important for the state and observation process,” with modifications to address the covariance among our predictor variables. We now describe our approach as a two-step model development process, and we tabulate the focal models.

Although I like the idea of figure 2 and I am aware that the information is in the text, the model of temporal trends was very difficult to follow. Writing down the formulation of the hierarchical model and including covariate names on table 3 would make the reading more straightforward.

Response: We have simplified Fig. 2 (omitting 2 nodes) and completely revised the text to more intuitively explain the hierarchical model, and revised the tables to list the candidate covariates in each sub-model.

Regarding gam models, I think they may not be correct since, if I understood correctly, for vegetation surveys only data from 2004, 2009, 2015 are available. If so, these models may not provide trustable information for vegetation surveys since only three points are available. As pointed out by referee 2, I suggest authors to remove these analyses and focus on the analyses of fig. 2. 

Response: We have relocated these graphs to an appendix more suitable for exploratory data visualization, and we have clarified in Methods and Results that we used generalized additive models only to visualize the pattern of occurrence of nutcrackers with respect to various potential covariates for which we had data from every nutcracker point-count station in every year, such as elevation or precipitation-as-snow. 

In addition, since covariates of white bark pines were predicted using data of these three surveys. It would be interesting to perform posterior predictive checks of the model, just to ensure that increased uncertainity throughout the model dit not lead to unreliable results

Response: Our previous draft included posterior predictive checks at two levels of the hierarchical model, and we have now added a third check for the whitebark sub-model, which can be checked at least against a few years of data. 

Regarding results, there are many figures some of them with similar information that could be reduced (i.e. fig. 7). 

Response: Each figure now contains only unique information; for example, nutcracker trends that appeared for reference as an overlay in several panels of Fig. 5 have been removed, and Fig. 7 has been moved to an appendix. 

I also agree with both referees that including a summary table of the results will make the manuscript more attractive.

Response: Summary tables of modeling results have been added for both the spatial and temporal models.

Finally, the description of the surveys and the study sites was difficult to follow because too much information was provided (see minor comments below) and most importantly, in some parts it was mixed with the “experimental design”. For example, until data-analysis section I was wondering if the plots of vegetation and transects of avian surveys were in the same area. I still have this doubt for the NCCN dataset. In addition, I am not sure that 7 minute surveys can be used as different visits. Aren’t they highly correlated? In sum, please separate well the description of the study area and the surveys. Clarify if vegetation plots and avian surveys were located in the same area for NCCN dataset. If not, please clarify how can the two processes be linked and modeled simultaneously.

Response: Each of these issues has been clarified in the revised text and figures. Text and references have been added to Methods to address the utility of 7-minute counts, in which data from sequential intervals of 3, 2 and 2 minutes are used to estimate bird availability for detection, and are not interpreted as independent visits. Figure 1 has been revised to show an example of the distribution of point-count stations relative to tree survey plots in a park. The extreme vagility of nutcrackers is explained to justify modeling annual changes in nutcracker density at the scale of whole parks, rather than at the scale of bird point-count stations or tree survey plots.

Overall I liked the idea of the work. It is clear that the authors have vast knowledge of the pine-nutcracker mutualism and that they are able to apply complex statistical analyses. Nonetheless, I had some doubts about model specifications, which made me wonder if the results were only reflecting type I errors, and if the dataset was adequate for their aims. If these issues are clarified or solved, I believe the work can make important contributions for our understanding of the nutcracker-white pine mutualism and its conservation.

Response: We appreciate your favorable review.

Additional Editor Comments:

Minor comments

L36-39. My suggestion is to use less technical words in the abstract.

Response: The abstract has been revised to remove technical jargon, such as “trophic”, “hierarchical N-mixture model” and “propagation of parameter uncertainty”.

L56-58 An example with plant-animal mutualisms would be better.

Response: We have added several examples of plant-animal mutualism.

L102-109 Please summarize

Response: We revised this section to emphasize that whitebark and nutcracker monitoring programs are unrelated, and a summary of our approach appears in the paragraph that follows L102-109.

Fig. 2 I have some doubts about how the processes are connected. For instance the probability that a nutcracker is available for detection does not influence the predicted number of nutcrackers (n) but the observed data (y).

Response: We have completely revised the text in this section to provide a more intuitive explanation of the processes being modeled. At the core of this new explanation are two simple equations, N*p[a]=n and n*p[d]=y, where N = true number of nutcrackers, p[a] = probability that a nutcracker is available for detection, n = true number of birds available for detection, p[d] = probability that a nutcracker is actually detected, and y = number of birds detected (the actual count). We use these equations to show that the probability that a nutcracker is available for detection, p[a], times the true number of nutcrackers, N, gives us the number of nutcrackers available for detection, n. Also, n times p[d], the probability that a nutcracker is actually detected, determines the observed count, y. We have also simplified the figure.

L123-126 I suggest to finish the introduction summarizing the aims of the studies and how they were approached, rather than with a technical description of the model.

Response: The introduction has been revised as suggested; see especially the final paragraph of the introduction for these revisions.

L137-138 Too much detail.

Response: These details of climate have been removed.

Fig. 1 I would add an explanatory figure of the sampling design. As another pannel of this one.

Table 1-2 I would move them to a supplementary material.

Response: Figure 1 has been revised as requested, and now includes an example of the non-overlapping distribution of whitebark and nutcracker plots in one park. Plot location details cannot be shown for all parks within a single figure, because plot symbols overlap when park images are small. 

L180 I guess panel is the altitude but it was confusing to me.

Response: Panels are random groupings of point-count stations, with each panel including a balanced representation of stations at higher and lower elevations. Each year, all stations in panel 1 (the “annual” panel) are visited, along with all stations in one of the other panels (the “alternate” panels). Alternate panels are visited in sequence in subsequent years. The text has been revised to clarify that in year 1 we survey stations in panels 1 and 2; in year 2 we survey stations in panels 1 and 3, and so on. This design allows us to visit a reasonable number of stations each year, while the number of stations visited over the lifetime of the project is much higher.

L182-183 Please clarify

Response: The text has been expanded to clarify that random transect locations were stratified by elevation.

L191 Please state that these intervals are used as “visits” for imperfect detection. Justify why potentially autocorrelation effects can be disregarded.

Response: Text and references have been added to address the utility of 7-minute counts, in which data from sequential intervals of 3, 2 and 2 minutes are used to estimate bird availability for detection, and are not interpreted as independent visits.

L194 How was ambient noise measured?

Response: Text was added to explain that observers categorized noise based on a scale (1-5) learned during training.

L196 How was aspect measured?

Response: Text was added to explain that aspect was determined using a digital elevation model based on point-count station coordinates. 

L207-208 At the end, it was not clear to me which variables were used. Please clarify and summarize your climatic variables.

Response: Tables were added to list and define focal variables in our final models.

L213-216 This is too vague. What is the particular suggestion of Bivand?

Response: This method has been replaced by a new spatial model.

L221-229 I thought gams were used for selecting covariates, but then they are presented as results. Nonetheless, I am not sure that they are correct.

Response: Methods and Results sections have been revised to clarify that these “models” were used only for data visualization, as a sort of data summary, similar to the way a mean and 95% confidence interval represent a model and summary of the data.

L244 Distance sampling was not included in the description of the survey.

Response: The original reference to distance sampling on line 191 has been expanded to emphasize this important component of our sampling design.

Table 3. Why the probability of perception is modeled with a log-link instead of a logit link?

Response: Because perceptibility declines continuously with distance, it is commonly modeled as a half-normal function of distance, where the shape parameter (sigma) controlling the half-normal is also a continuous random variable controlling the width of the half-normal curve. A logit link would not be appropriate because this is not a binomial process. Appropriate references receive more emphasis in the revised ms. 

L287 First time that canker is stated, please specify this in the survey, and describe it for non-specialized readers.

Response: The original reference to cankers on line 146 is now accompanied by a general description of cankers and associated signs of blister rust.

L289 Why the proportion of trees with cones was not included?

Response: Cones were detected only rarely in the data from NCCN parks, perhaps because whitebark did not mast during the 4-5 years of sampling in each park. This point is mentioned in the discussion and is now used to make a recommendation in the discussion.

L316 Please specify the value of the precision parameter and the value of the priors.

Response: We have provided a more complete description of priors and precision. 

L340-341 Please move this information to the data description. I was wondering this for a long time.

Response: The text has been revised as requested. We regret the omission of this important point during earlier editing of the ms.

L349 None of these analyses support the discussion and conclusions of the spatial relationships between pines and birds.

Response: We have removed these analyses from the ms. 

Table 4 Please move to an appendix

Response: We have removed this table from the ms, summarizing the few key statistics in line.

L388. Only three years, then I would not use year as a continuos covariate but as a factor.

L437 These estimates seem too big for a standardized covariate.

Response: We used data from 4 years of tree surveys in MORA (Table 1) to fit a linear model of each tree metric, and used this model to estimate all 13 years of data for each tree metric (i.e., we modeled crownkill per year based on crownkill data from 2004, 2007, 2009 and 2015, to estimate the annual mean and 95% CRI for crownkill in 2004-2016). We then used these modeled annual estimates of each whitebark metric (including the uncertainty in each annual estimate) as predictors of nutcracker density. This important feature of our model is now further emphasized in Methods and in Table 3.

Response: We have reviewed PLOS ONE style requirements and believe our ms meets all requirements.

2. In your Methods section, please provide additional location information of the study area, including geographic coordinates for the data set if available.

Response: We can provide point-count coordinates for the nearly 3000 bird monitoring stations used in this study, but we can only provide dithered coordinates (up to 140 m from actual coordinates) for the tree monitoring plots, because the exact plot locations are considered sensitive to protect the alpine-subalpine ecosystem in these parks. Given the questionable utility of dithered coordinates, and the large number of coordinates that would have to be reported, we have included only summary information on plot locations in the current ms. All plot locations are available through the National Park Service, and we will gladly consider additional options for reporting within this ms.

Response: We have added a “Permitting” statement to Methods, as requested.

Reviewers' comments:

Review #1: Ray et al. use data from four national parks in western North America to investigate the co-occurrence of whitebark pine (PIAL) and Clark’s nutcrackers (CLNU), as well as its temporal trends as the trees are declining due to blister rust. The data from the southern parks (SEKI; YOSE) shows that CLNU detections are more frequently detected in areas with pines, whether just PIAL (YOSE) or also foxtail pine (SEKI) The extensive analysis of time series from the two northern parks reveals that CLNU sightings are becoming less frequent in the Mount Rainier area (MORA), while they fluctuate strongly in the northern Cascades (NOCA). Based on the results, the authors then discuss that CLNU are flexible in their association with PIAL, as expected in this facultative mutualism, and that PIAL declines in the park may not affect the population sizes per se, but may force the birds to forage for their preferred seeds in other areas, outside of the managed NPS boundaries.

The paper is well written, addresses an important biological and conservation issue, and refrains from broad interpretations beyond the investigated phenomenon. Aside from some minor comments on the discussion, I have three suggestions aimed to improve the clarity of the article for the readers.

For full disclosure, I am not familiar with Bayesian modeling and could thus not assess whether the details of the analysis were performed correctly. The selection of variables, as well as the sampling regimens and data sources are satisfactory, however, and the authors have extensive experience and are well-regarded for these types of models. Therefore, I have no reason to doubt any of the methods or results presented.

- While co-occurrence, by definition, underlies the mutualism between the two species, it would be helpful to be clear about the fact that here, only co-occurrence is considered. Rather than speaking of the decline in the mutualism, it may be more accurate to speak of the “co-occurrence underlying the mutualism”, or “… that underlies the mutualism”, or something along those lines. This is just to point out that the actual movement ecology of foraging, seed caching, recovery, etc. is also involved in the mutualistic interaction, none of which is addressed by data in this MS.

Response: We appreciate the opportunity to rectify this oversight, and we have altered text in the introduction and added text in the discussion to acknowledge more clearly that our analysis addressed co-occurrence of nutcrackers and whitebark, a necessary but not sufficient condition for mutualism.

- A table that summarizes the findings from the complex model may be helpful. While most the findings are clearly communicated in the text and the figures, a simple table may help convey the wide array of variables considered and allows the authors to highlight those that were found to be important.

Response: We have added a table of model results, as requested.

- A minor wording issue, but “crown kill” makes a good proxy for potential seed production, but realized seed production can vary tremendously between years and cannot be captured with this metric.

Response: We have altered text in the introduction and added text to the discussion to further acknowledge that we have had to rely on proxies for seed production, which is not well characterized in our data.

- A stylistic issue which is really just a matter of preference: perhaps, the authors could consider flipping the color scheme in Figs. 3-7, so that the tree data is in grey-scale and the bird data contains color. Since the paper is more focused on the effects of tree demise on bird density, this would be better reflected and make it easier to interpret some of the more cluttered figures.

Response: We respectfully request to retain the color scheme we developed for the first draft of the ms, which was decided upon after much trial and error, and which we believe enhances the figures and facilitates their interpretation. Our focus is on tree metrics as a potential explanation of variation in nutcracker distribution and dynamics, which is one reason we highlighted tree metrics with color. Using color for tree metrics, rather than grey-scale, also allows us to distinguish between the several whitebark metrics without ambiguity.

- Finally, consider including the outline of the NCCN and SIEN in Fig. 1

Response: We thank the reviewer for this good suggestion. Because NCCN and SIEN have no official boundaries, we have differentiated networks by depicting SIEN parks in a lighter shade of grey within a revised version of Fig. 1.

Details:

- L 36: “trophic dependence” is not really what the models are investigating, it’s really about co-occurrence dynamics. No foraging data was collected.

Response: We have altered the text to clarify that co-occurrence is a suggestive but insufficient condition to infer trophic dependence.

- L 102: “Vital Signs”. Is this a NPS program? If so, consider describing it in short. The sudden capitalization is confusing.

Response: We have added a brief explanation of the NPS Vital Signs, as requested.

- L156: “4421 m in elevation”

- L164: “3,091 m in elevation”

Response: The elevational details on lines 156 and 164 were deleted in response to an editor’s request.

- Fig. 4a and b; confusing legend panel in Fig., as the tree species appears to be listed under CLNU as “detected”, “not detected” and “whitebark (or foxtail) pine” [consider removing last indentation]

Response: This issue was rectified in our new figure, which has also been updated with new spatial modeling results.

- Fig. 5: why are the CIs so small for 2009? Make sure to mention absence of CLNU in 2016 in MORA in caption

Response: Credible intervals were narrower in years when tree plots were actually surveyed, and were especially narrow in years like 2009 when all plots were surveyed. This explanation has been added to the figure caption, along with a reminder that nutcrackers were not detected in MORA in 2016.

- L 491: the data from SIEN almost seems like an afterthought in this paper. Have you considered showing the 6-year time series without modeling it? Otherwise, the paper may even benefit from dropping the SIEN part, since it does not tell us anything about temporal trends…

Response: By including the SIEN data at this early stage, we intended to motivate planning for more integrated analysis of NPS Vital Signs deriving from disparate datasets. Although it appears premature to model temporal dynamics based on just 6 years of data (during which most plots were surveyed only one or two times), we believed a space-for-time substitution approach could be useful as a first step toward inferring the potential for nutcracker dependence on tree resources. We agree, however, that our simple mapping of co-occurrence could be improved by developing a predictive model, and we have now developed a hierarchical model to predict nutcracker density as a function of whitebark and/or foxtail cover (derived from spatially complete vegetation maps) while accounting for imperfect detection of nutcrackers. Our new model suggests that the spatial distribution of nutcrackers is explained better in YOSE by whitebark cover than by elevation, and is explained best in SEKI by elevation, followed by whitebark cover and then foxtail cover. None of these results was particularly evident when viewing raw counts on our previous maps, so we thank the reviewers for the impetus to explore these patterns further. 

- L498-515: the two paragraphs are a bit contradictory, as you state in the first one that foxtail may attract CLNU which may then depredate PIAL seeds, and in the second that attracting to CLNY with foxtail may increase seed dispersal. This may be reconciled with some word-smithing.

Response: We have reversed the order of these paragraphs and revised the text to help clarify our argument that spillover of nutcrackers from foxtail into dwindling whitebark populations could either retard or accelerate whitebark decline. Foxtail presence might attract nutcrackers to areas with low whitebark numbers, perpetuating at least some seed dispersal, but low seed production might reduce the number of successful dispersal events below a threshold required for effective regeneration. 

- L553pp. Consider what may happen to the variability, not just the mean of nutcracker attendance. Perhaps they may just show up less frequently, or less reliably

Response: The first line in this paragraph refers to nutcracker dynamics in MORA, where both the mean and variance in nutcracker density appeared to decline over time, as shown in Fig. 6. We have altered the text, and the caption for Fig. 6, to clarify that we used our models to estimate the annual number of nutcrackers per hectare in MORA, rather than just mean density. The rest of this paragraph emphasizes the irruptive nature of nutcracker dynamics in NOCA and elsewhere throughout the range, and the ability of these birds to respond quickly to shifting resources, in order to provide context for the MORA result. Given the natural variation in nutcracker density, the MORA result appears to us all the more striking and suggestive that nutcrackers are shifting their activities outside this park in a deterministic manner.

- L592: here you state 650 miles, earlier they were kilometers with the same reference. Please reconcile

Response: Thank you for catching this typo; miles has been corrected to km.

- L 620: “proxies of potential cone production”

Response: The text has been altered as suggested.

Congratulations to the authors on a well-done and important study.

(PS - I assume the data will be made publicly available)

Response: We are thankful for your positive review. All data are made publicly available through the National Park Service.

Reviewer #2: Plos one review: Trends in the whitebark pine-Clark’s nutcracker mutualism

Thank you for the opportunity to review this manuscript. It was generally comprehensive and well-written with relevant conclusions to warrant publication in Plos one. I appreciate that the authors attempted to examine multiple facets of potential mutualism and association between nutcracker abundance and whitepine distribution, productivity, and pathogens. The integrated analysis examining the effects of whitepine trends on nutcracker abundance was particularly creative. My one general comment is that the paper is trying to do so much, it is quite difficult to follow through the methods and results. The authors are dealing with two primary analyses (temporal trend and spatial pattern) at 4 different sites, and each analysis has several different response variables and covariate sets. Instead of Figure 2, which provides a schematic of the N-mixture integrated model structure, perhaps a schematic of how each analysis or summary is structured would be much more helpful. Or a table of candidate models for each analysis. Something that structures that information and puts a reference in one place. Also, I suggest using additional sub-headings for different analyses or sites or something that helps readers index what information belongs to what. Other than restructuring the Methods/Results sections to provide additional clarity, the authors could include additional language describing model/covariate selection procedures. Otherwise, my comments are mostly editorial in nature (see below).

Response: We appreciate this generally favorable review and specific advice on potential improvements.

Line item comments:

Line 30-31: Remove scientific name from species because its in the abstract?

Line 64: Are names of species originators required as part of Plos one formatting? Suggest omitting here and elsewhere.

Response: We will gladly make these changes, pending advice from PLoS ONE.

Line 90-91: Suggest revising sentence to something more concise – for example, “Whitebark requires Clark’s nutcrackers to disperse its wingless seeds.

Response: Revised as suggested.

Line 111: This objective seems to incorporate only part of your analyses. The following paragraph (lines 117-126) delves more into methods, but doesn’t succinctly state the objective of each analysis – perhaps reword to focus more broadly on study objectives and reserve methodological discussion to the Methods or Discussion sections.

Response: Revised as requested. Study objectives have been stated more succinctly in this final paragraph of the introduction, a task facilitated by our new model of spatial variation in nutcracker density within SIEN parks, which parallels our temporal model of nutcracker density within NCCN parks, resulting in a more unified analysis that is easier to describe.

Line 176: Subheading for avian data?

Response: We have added subheadings for tree and avian surveys.

Line 178: Replace ‘to’ with ‘in’?

Response: This information was revised completely and moved up into the introduction for clarity.

Line 194-195: I’m not clear on what ‘cover density’ represents. I suggest briefly defining here and perhaps changing lower and higher cover to lower and higher density in parentheses?

Response: We have expanded our definition of dense cover as “presence of…any vegetation in which birds might escape detection.”

Line 211-220: Your spatial analyses and resulting figures seem a bit clunky/rudimentary. I’m confused why you reduced avian count data to apparent presence/absence despite having counts and information to estimate detection probability. If you used those data, you could create a nutcracker density map across SIEN sites that included whitebark or foxtail DBH, # of cone-producing trees, elevation, etc. as covariates. Summarized in grid cells that approximated the size of the area surveyed per point count. From this information you could provide helpful summary statistics such as XX% of the nutcracker population in SIEN occurred in whitebark dominated habitat, while XX% occurred in foxtail habitat. Or, cells with trees > XX DBH had XX% greater density of nutcrackers. Something more tangible than multiple pie charts over landcover layers that are difficult to interpret.

Response: We regret the clunky impression of our maps, which were intended as data visualizations rather than analyses. Originally, we did not pursue a spatial model linking nutcracker density to whitebark metrics because nutcracker point-count stations and tree survey plots were not co-located. However, we agree that estimating nutcracker density at each point is preferable to presenting only detection data, and our revised ms now presents spatial models of nutcracker density for SIEN parks. Our spatial model is similar to our temporal model except that we do not yet attempt to model whitebark metrics based on the currently sparse data from tree sampling plots. Instead, we relate nutcracker density to elevation and to the whitebark and foxtail pine cover maps used in our original ms. This approach can be adapted to include a model of whitebark distribution and/or dynamics as the tree monitoring database expands.

Lines 221-229: Did these avian exploratory analyses actually feed into any other models? It doesn’t appear so – I would delete this section and the corresponding section in the Results – find a way to incorporate the necessary covariates into a single temporal analysis.

Response: We have relocated these visualizations of the data to an appendix.

Line 231: Can you be more specific than whitebark dynamics? It would add clarity to define the response variables in the GLMs that fed into the N-mixture model as covariates.

Response: This section has been completely revised to provide a more intuitive and methodical explanation of our models, and we have expanded Tables 3 and 4 to list the specific whitebark metrics and other covariates used in our models of nutcracker density.

Table 3: Not very informative – W = whitebark metrics and x = other covariates. See general comment, but some detailed summary of what actually was examined in each model component would be useful here and for other analyses.

Response: We have expanded Tables 3 and 4, which now list the specific whitebark metrics and other covariates used in our models of nutcracker density.

Lines 297-300: I’m confused as to how you actually conducted model selection. Here you state you included each voariate in a set of candidate models with 0-4 additional covariates. Then on lines 323-325 you state you examined GOF statistics and overlapping 95% CIs, but if you included a varying number of covariates per model, you must have somehow decided which covariate combination was the ‘winner’ in order to interpret coefficients? I’m not sure if this was poorly done or I’m missing the authors’ intent. In the Results section, Bayes-P and LOF ratio are reported as if they provide evidence in support of a specific covariate set – which is an inappropriate use of those metrics. Regardless, if you have a candidate model set for each analysis, it would be worth including it as a table or appendix. Further, if you’re looking for a more formal way to conduct Bayesian model selection, there are few options including WAIC or LOO, which can be implemented pretty easily in R using the LOO package by exporting the log likelihood estimates from your jags model and running a couple lines of code. From that you can obtain an information criterion metric that could be used to formally compare models. Regardless, this section needs to be rewritten to facilitate understanding of the actual model/covariate selection process.

Response: We regret this confusion and have removed all model-selection terminology from the ms. We now cite papers suggesting that our models aren’t suited to WAIC or LOO, and that hierarchical mixed models with overdispersion are still a challenge for model selection (Hooten and Hobbs 2015, Williams et al. 2019, Plard et al. 2019). We focus instead on model development and exploration. As before, our focus is on a pair of models, one with and one without an effect of whitebark on nutcrackers (call them a and b) that make very similar predictions given the available data. As before, we still refrain from ranking these models, but now we describe how we selected the covariates within them. Briefly, we started with a relatively full model (barring correlated covariates) and used backward stepwise elimination of covariates while monitoring posterior predictive checks to halt eliminations before Bayesian P-values became too extreme (<0.2 or >0.8). We recognize that Bayesian P-values provide an optimistic metric of model fit, so we are careful not to over-interpret our results at this point, and to suggest a more rigorous model-selection framework going forward when there are enough data to support CRV, etc.

Line 316: Why mean = 1.0 here and not 0.0, which is more typical for uninformative random effect priors?

Response: Thank you for catching this error; our priors all had mean = 0, as is now stated correctly in the ms.

Line 385-388: So you’re really basing your temporal trends on three data points for whitebark metrics. Is the implication of such a minimal sample size on your ability to make inference warranted in the Discussion?

Response: Although a fourth year of partial sampling also contributed to our estimates of trend in MORA whitebark, we agree that it is important not to overstate the evidence for nutcracker dependence on whitebark. We have focused on model development rather than model selection in part because our data are still sparse for models of this complexity (e.g., k-fold cross-validation is at odds with model convergence). We have revised our discussion to further emphasize limitations of the current data.

Line 456-473: So, these results are from a series of ad hoc GAMs on the apparent presence/absence of nutcrackers? Why abandon the results of your N-mixture model to delve into covariate effects from this analysis? Could these covariates not have entered into your N-mixture model, and if they weren’t supported, shouldn’t be discussed further?

Response: We regret the original position of these plots within the ms. We did not actually proceed from N-mixture models to GAMs; instead, we used GAMs to illustrate, visually, relationships in the data that inspired our models. We have moved these plots to an appendix and now address them earlier in our results section, and use them to demonstrate our model-development process. We also include them to guide model construction in future analyses of data from these ongoing Vital Signs monitoring programs. 

Line 482: The Discussion is quite long and focuses extensively on scenarios that call their study into question such as dynamics outside of park boundaries and evidence suggesting nutcrackers aren’t that dependent upon whitebark. I think these sections are important to include but could be shortened significantly. Focus on the interesting results from your study, how they fit in the broader literature, what issues occurred with the data and your ability to make inference, and what could be done going forward.

Response: We have shortened several sections of the discussion, added our suggestion for inference going forward, and further clarified how this study fits into the broader literature. We also clarify that nutcrackers might be using whitebark resources outside the parks.

Line 624: All other citations are numbers, why author and year here?

 Response: Thank you. This oversight has been corrected.

---

## [Decision Letter · Decision Letter 1]

25 Jun 2020

PONE-D-19-34230R1

Assessing trends and vulnerabilities in the mutualism between whitebark pine (*Pinus albicaulis*) and Clark’s nutcracker (*Nucifraga columbiana*) in national parks of the Sierra-Cascade region

PLOS ONE

Dear Dr. Ray

Thank you for submitting your manuscript to PLOS ONE. After careful consideration, we feel that it has merit but does not fully meet PLOS ONE’s publication criteria as it currently stands. Therefore, we invite you to submit a revised version of the manuscript that addresses the points raised during the review process.

Thank you for such an extensive review of the previous version of the manuscript. I have found the material and methods clearer and more easy to follow. Now, I could understand the sampling protocol as well as the analyses performed and they are correct. All the issues raised by the first referee were solved and he thinks (as I do) that the contribution of this work is relevant.

 However, even though the methods section has been clarified I agree with the second referee that some parts are still difficult to follow.  I had to read Amudson’s work (2014) to fully understand fig. 2. A full formulation of the model will help readers to better understand your work. 

 In the current version it is difficult to connect all parts of the model.  For instance, eq 1 reflects the relationship between the probability of non-detection (q) and covariates, but the relationship between (q) and availability is not formulated. Similarly, in eq. 2 the relationship between the scale parameter of the half-normal distribution is stated, but the information about its relationship with perceptibility is scattered throughout the text (see comments below). Most importantly, the link between population parameters and observation processes was not totally connected even though its formulation would be quite simple. (1) A binomial distribution that links the number of individuals observed (y) to those available (n) with a probability of detection (pd). Pd, in turns depends on the effects of covariates on the scale parameter sigma. (2) A binomial distribution that connects the number of available individuals (n) with the actual number of individuals (N) with a probability of availability (pa). Pa depends on the probability of non-detection (q) which is modulated by certain covariates. (3) A poisson model that links the expected number of individuals (lambda) with environmental covariates. Stating these formulas in the text, together with fig. 2 would really help to understand your modeling approach.  

In sum, I congratulate the authors for the hard work and the revision of the previous version. I agree with the analyses performed but prior to publication the formulation of the model needs to be within the manuscript (if not in the main text in a supplementary material).  To make the reading more straightforward, I suggest to include the formulation in the main text while shortening some parts (i.e. L265-269).  

We look forward to receiving your revised manuscript.

Kind regards,

Teresa Morán-López

Academic Editor

PLOS ONE

Additional Editor Comments (if provided):

Minor comments

L105-120 this part gets a little bit to methodological in the introduction. I think an important contribution of this work is that the modeling approach allows to unify different monitoring protocols. Maybe some of the methodological parts can be summarized and your novel approach can be highlighted with references to other modeling efforts in the same line. This is only a suggestion

Table 1 heading. Please specify that these are NCNN parks and that this dataset will be used for the spatial analyses. For readers that are not familiar with the area there are many acronyms and it gets difficult to follow. The same applies for table 2.

L189 How many panels?

L204. Please briefly describe aspect for non-specialized readers.

L207-216 The way climatic covariables are explained is not clear. At the end of this paragraph I was not sure which variables were used. It was not until table 4 that I found out that it was PAS and MSTr. I also got lost with the time lag used (previous or current year) and how an anomaly was defined. I guess it is the MST residual. Please specify this and give a biological interpretation of these residuals. In sum, rewrite this paragraph making sure that the reader knows which climatic variables were used, their interpretation and time-lag .

L255 Figure 2 legend and along the text.  To make the formulation consistent with that of Amudsons’ I suggest to use the probability of detection instead (ak in Amudsons’ formulation).

L273 This would need a connection between the half-normal function and the probability of detection. I think that formulation would help. If not, information provided in L273 should be in L291-293 to make it easier to follow.

Table 5 In the spatial analyses were whitebark and foxtail parameters normalized? If I understood correctly in the temporal analyses they were not (because they were estimated). Please clarify this to make sure that comparisons between parameters strength are correct. 

Fig. 1B I could not see the dots. I guess they are lines.

Reviewers' comments:

Reviewer's Responses to Questions

**Comments to the Author**

1. If the authors have adequately addressed your comments raised in a previous round of review and you feel that this manuscript is now acceptable for publication, you may indicate that here to bypass the “Comments to the Author” section, enter your conflict of interest statement in the “Confidential to Editor” section, and submit your "Accept" recommendation.

Reviewer #1: All comments have been addressed

Reviewer #3: (No Response)

2. Is the manuscript technically sound, and do the data support the conclusions?

Reviewer #1: Yes

Reviewer #3: Partly

3. Has the statistical analysis been performed appropriately and rigorously? 

Reviewer #1: Yes

Reviewer #3: I Don't Know

4. Have the authors made all data underlying the findings in their manuscript fully available?

Reviewer #1: Yes

Reviewer #3: No

5. Is the manuscript presented in an intelligible fashion and written in standard English?

Reviewer #1: Yes

Reviewer #3: Yes

6. Review Comments to the Author

Reviewer #1: The authors have done a great job revising the manuscript following extensive comments by the reviewers and editors. To me, the most interesting finding is that YEAR is the strongest predictor of the observed decline in CLNU detections, strongly suggesting that in addition to declines in food species, other factors may contribute to population dynamics in this species. Overall, this is an important contribution and I look forward to seeing future results from this modeling approach

Reviewer #3: Thanks for the opportunity to review this paper. The subject matter is interesting, and the introduction and discussion were generally clear, interesting, and well written. The methods, and subsequently results however need substantial work. Following this, the work should be a valuable contribution to the scientific literature.

The main issue is that the methods do not clearly explain exactly what models were fit and how. I believe that there are two overarching problems.

The first is linguistic: The authors fit hierarchical models, which contain many parts. In the description of these parts, the authors often refers to these parts (which are generally link functions or sub-models) as "models" or "GLMs". This is incorrect and very confusing.

As far as I can tell (and this relates to the following problem) the models were fit as a hierarchical model, e.g. JAGS models supp 2. The frequent reference to parts of these hierarchical models as "models" themselves is very confusing. E.g. lines 400-430.

The equations showing the link functions 1, 2, and 3, on lines 433, 437, and 452 are equations, and should be labelled as such. However these are later referred to in text as "models" (lines 462, 468).

Please clarify this section.

The second problem is that I cannot understand how many models were fit and which they were. This is compounded by the first problem which throws the word 'model' around when it should not be.

Tables 5 and 6 show a series of models. I don't understand how you used backwards selection from a full model to arrive at different these different models - shouldn't you arrive at a single model? Further (t5) I don't understand how you eliminated various versions of Wk (t3) to choose whitebark cover as the measure. It may well be explained, but it's not clear to me where from careful reading of the methods and supps. This must be clarified.

In addition to working on your explanation, I suggest three key structural things would go a long way to helping resolve this:

Firstly make a full description of your model specifications as equations. At present only parts of the model are shown, i.e., in equations 1-3. Put the whole model down, either in text or supp. The JAGS code and DAG are only useful for people who already know JAGS and are used to interpreting DAGs. E.g.

N ~ Poisson(lambda)

y = pd x pa x N etc.

Secondly include a complete table of all models fit and any relevant parameters involved in the selection process (e.g. bayes p value), probably in the supplementary material.

Thirdly, the code supplied does not meet modern standards for reproducibility, and it should.

At present JAGS model code is included in supp 2. There is no R code but a wrapper around the model. There is no data provided.

Although the authors state in the declaration to the journal that "all data are fully available without restriction", this does not appear to be the case. The description of where the data may be found points to a series of reports with summaries of the data, not the data themselves. It is not clear where the raw data may be obtained of if these truly are publicly available.

This lack of code and data does not allow others to understand the analysis, let alone reproduce it. I strongly recommend that the raw data be made available if they are not, and that full analysis code be made available and directly linked to it. This may make use of a public code repository such as github, in preference to more code in a word document. Yes this is more work, but it is current good practice and means that anyone should be able to follow your analysis.

I cannot stress enough that the methods section is at present very difficult to follow and this must be addressed prior to considering publication. It appears that previous editors and reviewers have been unable to clearly understand what you have done, and this is certainly the case for me. Most readers will not pay nearly as much attention, and so you must make significant efforts to ensure that your work is clear and understandable to even a casual reader.

I have a third concern with this study: that the shift from surveying lower to higher elevations later in the year following nutcracker movements is likely to overestimate landscape densities if individuals may be counted at one low elevation site early in the year and other higher elevation sites afterwards. This confounding in sampling means that estimates of N will only be relevant to that precise time, not a general estimate of nutcracker abundance. This point should be addressed in the discussion, and preferrably in future sampling. This is a problem for the generalisability of the conclusions drawn from this sampling.

A couple of other minor points:

Abstract: The place name acronyms make the abstract harder to read, not easier. Please don't punish your readers by making them recall four novel acronyms in the abstract.'

Table 1 & 2: subscripts a and b are unnecessary - just change "survey period" to "survey months" and state in table label tha count of trees includes live and dead trees.

L230. Bird surveys?

7. PLOS authors have the option to publish the peer review history of their article (what does this mean?). If published, this will include your full peer review and any attached files.

Reviewer #1: **Yes: **Mario Pesendorfer

Reviewer #3: No

---

## [Author Response · Author response to Decision Letter 1]

24 Aug 2020

PONE-D-19-34230R1

Assessing trends and vulnerabilities in the mutualism between whitebark pine (Pinus albicaulis) and Clark’s nutcracker (Nucifraga columbiana) in national parks of the Sierra-Cascade region

PLOS ONE

RESPONSE TO REVIEW (see RESPONSEs below)

Thank you for such an extensive [revision] of the previous version of the manuscript. I have found the material and methods clearer and more easy to follow. Now, I could understand the sampling protocol as well as the analyses performed and they are correct. All the issues raised by the first referee were solved and he thinks (as I do) that the contribution of this work is relevant.

 However, even though the methods section has been clarified I agree with the second referee that some parts are still difficult to follow. I had to read Amudson’s work (2014) to fully understand fig. 2. A full formulation of the model will help readers to better understand your work. 

 In the current version it is difficult to connect all parts of the model. For instance, eq 1 reflects the relationship between the probability of non-detection (q) and covariates, but the relationship between (q) and availability is not formulated. Similarly, in eq. 2 the relationship between the scale parameter of the half-normal distribution is stated, but the information about its relationship with perceptibility is scattered throughout the text (see comments below). Most importantly, the link between population parameters and observation processes was not totally connected even though its formulation would be quite simple. (1) A binomial distribution that links the number of individuals observed (y) to those available (n) with a probability of detection (pd). Pd, in turns depends on the effects of covariates on the scale parameter sigma. (2) A binomial distribution that connects the number of available individuals (n) with the actual number of individuals (N) with a probability of availability (pa). Pa depends on the probability of non-detection (q) which is modulated by certain covariates. (3) A poisson model that links the expected number of individuals (lambda) with environmental covariates. Stating these formulas in the text, together with fig. 2 would really help to understand your modeling approach. 

RESPONSE: Thank you for the favorable review and detailed suggestion for revision. We have implemented this particular revision within the first paragraph of Methods / Analyses.

In sum, I congratulate the authors for the hard work and the revision of the previous version. I agree with the analyses performed but prior to publication the formulation of the model needs to be within the manuscript (if not in the main text in a supplementary material). To make the reading more straightforward, I suggest to include the formulation in the main text while shortening some parts (i.e. L265-269). 

RESPONSE: We have included text on the model formulation in the main text and have shortened the indicated section as suggested.

Minor comments

L105-120 this part gets a little bit to methodological in the introduction. I think an important contribution of this work is that the modeling approach allows to unify different monitoring protocols. Maybe some of the methodological parts can be summarized and your novel approach can be highlighted with references to other modeling efforts in the same line. This is only a suggestion

RESPONSE: To reduce the methodological detail in this paragraph, we moved one sentence (with details on the dates of survey establishment in each network) into a later paragraph. We opted to retain most other elements of this paragraph in order to support our argument that a unifying model is appropriate and necessary.

Table 1 heading. Please specify that these are NCNN parks and that this dataset will be used for the spatial analyses. For readers that are not familiar with the area there are many acronyms and it gets difficult to follow. The same applies for table 2.

RESPONSE: We made the requested changes to captions of Tables 1 and 2.

L189 How many panels?

RESPONSE: We inserted the number of panels (six).

L204. Please briefly describe aspect for non-specialized readers.

RESPONSE: We have added text to provide guidance for readers.

L207-216 The way climatic covariables are explained is not clear. At the end of this paragraph I was not sure which variables were used. It was not until table 4 that I found out that it was PAS and MSTr. I also got lost with the time lag used (previous or current year) and how an anomaly was defined. I guess it is the MST residual. Please specify this and give a biological interpretation of these residuals. In sum, rewrite this paragraph making sure that the reader knows which climatic variables were used, their interpretation and time-lag .

RESPONSE: We have added text to explain the biological justification for the time-lag we considered. We have also clarified why we used residuals of MST on PAS in our models, citing a paper for this approach, which is adopted simply to avoid collinearity due to correlation between MST and PAS.

L255 Figure 2 legend and along the text. To make the formulation consistent with that of Amudsons’ I suggest to use the probability of detection instead (ak in Amudsons’ formulation).

RESPONSE: We use q, the per-minute probability of non-detection, because q was referenced in both the original model of availability by Farnsworth et al. (2002) and by Amundson et al. (2014; e.g., page 480), and because the code we have previously published (Ray et al. 2017) is based on q. For these reasons, we are requesting permission to retain q as the variable of interest, but we have attempted to explain q more clearly by adding “q = 1-a” in this clause.

L273 This would need a connection between the half-normal function and the probability of detection. I think that formulation would help. If not, information provided in L273 should be in L291-293 to make it easier to follow.

RESPONSE: We have revised this text as “We then used the common approach of approximating the decline in detection probability with distance from the observer using a half-normal distribution fitted to b, our (binned) data on distance-to-detection. Using this approach, pd was a function of σ, the fitted scale parameter of the half-normal distribution [39, 51, 59].”

Table 5 In the spatial analyses were whitebark and foxtail parameters normalized? If I understood correctly in the temporal analyses they were not (because they were estimated). Please clarify this to make sure that comparisons between parameters strength are correct. 

RESPONSE: Yes, whitebark and foxtail cover were standardized in the spatial analysis. We have now clarified this point. In the temporal analysis, which used data from NCCN parks, there were no foxtail present, so we did not compare relative effects of whitebark and foxtail on nutcracker dynamics. 

Fig. 1B I could not see the dots. I guess they are lines.

RESPONSE: Yes, the dots representing individual point-count stations appear to merge into lines along each transect, so we have revised the text to better describe the figure. Thank you for your attention to detail.

Reviewers' comments:

Reviewer's Responses to Questions

Comments to the Author

1. If the authors have adequately addressed your comments raised in a previous round of review and you feel that this manuscript is now acceptable for publication, you may indicate that here to bypass the “Comments to the Author” section, enter your conflict of interest statement in the “Confidential to Editor” section, and submit your "Accept" recommendation.

Reviewer #1: All comments have been addressed

Reviewer #1: The authors have done a great job revising the manuscript following extensive comments by the reviewers and editors. To me, the most interesting finding is that YEAR is the strongest predictor of the observed decline in CLNU detections, strongly suggesting that in addition to declines in food species, other factors may contribute to population dynamics in this species. Overall, this is an important contribution and I look forward to seeing future results from this modeling approach

RESPONSE: Thank you for your favorable review. We have emphasized the effect of year as suggested, especially in the 5th paragraph of the discussion.

Reviewer #3: Thanks for the opportunity to review this paper. The subject matter is interesting, and the introduction and discussion were generally clear, interesting, and well written. The methods, and subsequently results however need substantial work. Following this, the work should be a valuable contribution to the scientific literature.

The main issue is that the methods do not clearly explain exactly what models were fit and how. I believe that there are two overarching problems.

The first is linguistic: The authors fit hierarchical models, which contain many parts. In the description of these parts, the authors often refers to these parts (which are generally link functions or sub-models) as "models" or "GLMs". This is incorrect and very confusing.

As far as I can tell (and this relates to the following problem) the models were fit as a hierarchical model, e.g. JAGS models supp 2. The frequent reference to parts of these hierarchical models as "models" themselves is very confusing. E.g. lines 400-430.

The equations showing the link functions 1, 2, and 3, on lines 433, 437, and 452 are equations, and should be labelled as such. However these are later referred to in text as "models" (lines 462, 468).

Please clarify this section.

RESPONSE: We thank the reviewer for his fair assessment that our terms might prove confusing. We have replaced several instances of “model” with “sub-model” and have added descriptive adjectives (e.g., “observation sub-model”) where necessary. However, we note that many papers on hierarchical models have referred to these sub-models as models and as GLMs. For example, Amundsen et al. (2014) state, “Etterson et al. (2009) created a similar hierarchical N-mixture model using Farnsworth et al.’s (2002) time-removal method to estimate abundance adjusted for pa. In these models, covariate effects on both abundance and detection can be modeled directly as a generalized linear model.”

The second problem is that I cannot understand how many models were fit and which they were. This is compounded by the first problem which throws the word 'model' around when it should not be.

Tables 5 and 6 show a series of models. I don't understand how you used backwards selection from a full model to arrive at different these different models - shouldn't you arrive at a single model? Further (t5) I don't understand how you eliminated various versions of Wk (t3) to choose whitebark cover as the measure. It may well be explained, but it's not clear to me where from careful reading of the methods and supps. This must be clarified.

In addition to working on your explanation, I suggest three key structural things would go a long way to helping resolve this:

Firstly make a full description of your model specifications as equations. At present only parts of the model are shown, i.e., in equations 1-3. Put the whole model down, either in text or supp. The JAGS code and DAG are only useful for people who already know JAGS and are used to interpreting DAGs. E.g.

N ~ Poisson(lambda)

y = pd x pa x N etc.

RESPONSE: We have added text in the first paragraph of Methods / Analyses that provides an overview of the hierarchical model and the links between the population and observation sub-models.

Secondly include a complete table of all models fit and any relevant parameters involved in the selection process (e.g. bayes p value), probably in the supplementary material.

RESPONSE: We have described our full model development procedure, including all fitted models and selection criteria in Supplement S2.

Thirdly, the code supplied does not meet modern standards for reproducibility, and it should.

At present JAGS model code is included in supp 2. There is no R code but a wrapper around the model. There is no data provided.

Although the authors state in the declaration to the journal that "all data are fully available without restriction", this does not appear to be the case. The description of where the data may be found points to a series of reports with summaries of the data, not the data themselves. It is not clear where the raw data may be obtained of if these truly are publicly available.

This lack of code and data does not allow others to understand the analysis, let alone reproduce it. I strongly recommend that the raw data be made available if they are not, and that full analysis code be made available and directly linked to it. This may make use of a public code repository such as github, in preference to more code in a word document. Yes this is more work, but it is current good practice and means that anyone should be able to follow your analysis.

RESPONSE: We have cleared the release of all data from the National Park Service, and have provided new links to these data in Supplement S2. We have also previously published all the R code required to process the nutcracker data (cited as [54]).

I cannot stress enough that the methods section is at present very difficult to follow and this must be addressed prior to considering publication. It appears that previous editors and reviewers have been unable to clearly understand what you have done, and this is certainly the case for me. Most readers will not pay nearly as much attention, and so you must make significant efforts to ensure that your work is clear and understandable to even a casual reader.

RESPONSE: We have added text in the first paragraph of Methods / Analyses that provides an overview of the hierarchical model and the links between the population and observation sub-models.

I have a third concern with this study: that the shift from surveying lower to higher elevations later in the year following nutcracker movements is likely to overestimate landscape densities if individuals may be counted at one low elevation site early in the year and other higher elevation sites afterwards. This confounding in sampling means that estimates of N will only be relevant to that precise time, not a general estimate of nutcracker abundance. This point should be addressed in the discussion, and preferrably in future sampling. This is a problem for the generalisability of the conclusions drawn from this sampling.

RESPONSE: We note that our temporal analysis focused on Mount Rainier National Park, where nutcrackers were detected only at high-elevations, so the particular effect that you (rightly) warn of is not relevant to that analysis. However, our other analyses might overestimate nutcracker density for the reason suggested, and we acknowledge this and other limitations to the current temporal aspects of bird sampling design. Our discussion includes, “seasonal variation in resource use requires that we match dates of nutcracker monitoring to the dates of resource use. The timing of standard breeding-bird point-counts might not coincide with the season(s) in which nutcrackers are most frequent or detectable in these parks. Monitoring in the late summer and fall should offer the best opportunity to detect nutcracker use of mature five-needle pine seeds.” 

A couple of other minor points:

Abstract: The place name acronyms make the abstract harder to read, not easier. Please don't punish your readers by making them recall four novel acronyms in the abstract.'

RESPONSE: We have removed these acronyms from the abstract.

Table 1 & 2: subscripts a and b are unnecessary - just change "survey period" to "survey months" and state in table label tha count of trees includes live and dead trees.

RESPONSE: We have reduced the number of subscripts using the first change suggested. We retained subscripts that provide key details about the counts of live and dead trees.

L230. Bird surveys?

RESPONSE: We assume this comment is meant to suggest that bird surveys should be addressed in this paragraph. The paragraph has been expanded with a brief description of the model structure, including elements parameterized using data from bird surveys.

---

## [Editor Report · Decision Letter 2]

2 Sep 2020

Assessing trends and vulnerabilities in the mutualism between whitebark pine (*Pinus albicaulis*) and Clark’s nutcracker (*Nucifraga columbiana*) in national parks of the Sierra-Cascade region

PONE-D-19-34230R2

Dear Dr. Ray,

We’re pleased to inform you that your manuscript has been judged scientifically suitable for publication and will be formally accepted for publication once it meets all outstanding technical requirements.

Kind regards,

Teresa Morán-López

Academic Editor

PLOS ONE
---

## [Editor Report · Acceptance letter]

22 Sep 2020

PONE-D-19-34230R2 

Assessing trends and vulnerabilities in the mutualism between whitebark pine *(Pinus albicaulis)* and Clark’s nutcracker *(Nucifraga columbiana)* in national parks of the Sierra-Cascade region 

Dear Dr. Ray:

I'm pleased to inform you that your manuscript has been deemed suitable for publication in PLOS ONE. Congratulations! Your manuscript is now with our production department. 

Kind regards, 

on behalf of

Dr. Teresa Morán-López 

Academic Editor

PLOS ONE